# Desensitisation of Notch signalling through dynamic adaptation in the nucleus

Ranjith Viswanathan[1,†], Jonas Hartmann[1,2,*,†] (iD), Cristina Pallares Cartes[1] & Stefano De Renzis[1,**] (iD)

## Abstract

During embryonic development, signalling pathways orchestrate organogenesis by controlling tissue-specific gene expression programmes and differentiation. Although the molecular components of many common developmental signalling systems are known, our current understanding of how signalling inputs are translated into gene expression outputs in real-time is limited. Here we employ optogenetics to control the activation of Notch signalling during *Drosophila* embryogenesis with minute accuracy and follow target gene expression by quantitative live imaging. Light-induced nuclear translocation of the Notch Intracellular Domain (NICD) causes a rapid activation of target mRNA expression. However, target gene transcription gradually decays over time despite continuous photo-activation and nuclear NICD accumulation, indicating dynamic adaptation to the signalling input. Using mathematical modelling and molecular perturbations, we show that this adaptive transcriptional response fits to known motifs capable of generating near-perfect adaptation and can be best explained by state-dependent inactivation at the target *cis*-regulatory region. Taken together, our results reveal dynamic nuclear adaptation as a novel mechanism controlling Notch signalling output during tissue differentiation.

**Keywords** near-perfect adaptation; Notch signalling; optogenetics; signalling adaptation; transcriptional regulation
**Subject Categories** Computational Biology; Development; Signal Transduction
**The EMBO Journal (2021) 40: e107245**

## Introduction

Signalling dynamics have emerged as fundamental to cellular information processing in the context of tissue differentiation and organogenesis (Freeman & Gurdon, 2002; Sonnen & Aulehla, 2014). The cellular response to a signalling stimulus depends on the concentration, duration and frequency at which cells are exposed to it (Rogers & Schier, 2011; Mc Mahon, Sim *et al*, 2014; Johnson & Toettcher, 2019; Keenan *et al*, 2020). However, understanding how cells integrate and translate dynamic signalling information into specific gene expression outputs *in vivo* has proven challenging, as it requires methods for precise spatio-temporal control of the input and quantification of output levels in real time (Sako *et al*, 2016; Huang *et al*, 2017; Krueger *et al*, 2019; McDaniel *et al*, 2019; Hartmann *et al*, 2020; Johnson *et al*, 2020; Rogers & Müller, 2020). Towards this aim, we combined optogenetics with live imaging of nascent transcripts to study the input–output relationship linking Notch signalling activation and target gene expression during early *Drosophila* embryogenesis.

Notch signalling is a conserved pathway controlling cell fate decisions through direct cell–cell contacts (Artavanis-Tsakonas *et al*, 1999; Bray, 2016). Upon binding to one of its ligands (Delta or Serrate in *Drosophila*) presented on the surface of an adjacent signal-sending cell, the Notch receptor undergoes a series of intra-membranous proteolytic cleavages culminating in the release of NICD, a soluble cytosolic fragment. Cleaved NICD translocates into the nucleus, where it activates target gene expression by interacting with the DNA-binding protein Suppressor of Hairless Su(H) (Bray & Furriols, 2001; Gomez-Lamarca *et al*, 2018). As such, Notch signalling does not involve any signal amplification step, and its strength is thought to be directly related to the level of NICD production.

Our findings reveal an unexpected dynamic input–output relationship between Notch signalling and target gene expression. Optogenetically induced nuclear translocation of NICD causes rapid induction of target mRNA expression followed by gradual inactivation over time, despite continuous photo-activation and nuclear NICD accumulation. Such desensitisation, where a system returns to basal or near-basal levels of activity despite continued stimulation, is known as perfect or near-perfect adaptation (Barkai & Leibler, 1997; Ma *et al*, 2009; Sorre *et al*, 2014; Ferrell, 2016). By combining mathematical modelling and molecular perturbations, we show that the adaptive transcriptional response to Notch signalling is best explained by a known fundamental regulatory motif that implements adaptation through state-dependent inactivation (Friedlander & Brenner, 2009).

Adaptation can serve different functions, including limiting the duration of a signalling response and allowing cells to detect changes

1 European Molecular Biology Laboratory, Developmental Biology Unit, Heidelberg, Germany
2 Department of Cell and Developmental Biology, University College London, London, UK
*Corresponding author. Tel: +44 784 289 2414; E-mail: jonas.m.hartmann@protonmail.com
**Corresponding author. Tel: +49 6221 387 8109; Fax: +49 6221 387 8166; E-mail: derenzis@embl.de
†These authors contributed equally to this work

in molecules over a broad range of concentration. Well-established examples of adaptation involve regulation of signal transduction components at the cell membrane or in the cytoplasm, as in the case of EGFR (Vieira *et al*, 1996) or chemokine receptor downregulation following ligand stimulation (Yuan *et al*, 2012). Our results demonstrate that even for a simple signal transduction system, where a cleaved surface receptor directly controls target gene expression, dynamic and context-sensitive input–output relationships can be implemented at the level of transcriptional regulation.

# Results

### Optogenetic control of NICD nuclear translocation and target gene expression

To gain direct and time-resolved control over Notch signalling, we constructed an optogenetic system that allows reversible induction of NICD nuclear translocation by light (opto-Notch). To minimise dark state activity (i.e. NICD nuclear translocation in the dark), we designed a double-gated strategy by coupling NICD nuclear localisation signal (NLS) photo-tagging with dark state-specific mitochondrial trapping. We employed LANS (Niopek *et al*, 2014; Guntas *et al*, 2015) to cage the NLS of NICD with a photo-sensitive LOV domain and LOVTRAP (Wang *et al*, 2016) to anchor NICD-LOV to mitochondria through a Zdark tag, which recognises the dark conformation of the LOV domain and was itself localised to mitochondria by a TOM20 tag. Blue light illumination should cause simultaneous unbinding from mitochondria and uncaging of the NLS, resulting in NICD nuclear translocation (Fig 1A and B). Embryos co-expressing the opto-Notch module (with NICD tagged also with mCherry to follow its localisation) demonstrate that in the dark NICD is localised to particulate structures in the cytoplasm (Fig 1C and Movie EV1) and upon blue light illumination ($\lambda = 488$ nm), it rapidly translocated into embryonic nuclei, reaching maximum accumulation after 10 min (Figs 1D and EV1A, and Movie EV1). To test whether NICD nuclear translocation was functional and could elicit a signalling response, we measured target gene expression by quantitative live imaging of *sim* nascent transcripts using the MS2-MCP system (Garcia *et al*, 2013). *sim* is a well-established Notch target that specifies the embryonic mesectoderm (Bardin & Schweisguth, 2006; De Renzis *et al*, 2006; Falo-Sanjuan *et al*, 2019; Viswanathan *et al*, 2019). In addition to Notch, its expression requires input from the maternal Dorsal gradient and zygotic expression of the transcription factor Twist (Kasai *et al*, 1998). In the mesoderm, *sim* expression is repressed by the transcription factor Snail (Cowden & Levine, 2002). As a result of this regulation, *sim* expression starts towards the end of cellularisation, before the onset of gastrulation in a single row of cells on either side of the mesoderm (Fig 1E). Optogenetic activation at the beginning of cellularisation caused a premature activation of *sim* transcription, appearing as a bright nuclear dot from ~5 min after NICD began translocating into the nucleus (Fig 1F and G, Movie EV2). Importantly, *sim* was not transcribed in the dark at this stage (Fig 1F), demonstrating the validity of our approach. Consistent with the regulatory network controlling *sim* expression, optogenetic activation did not cause *sim* expression in the mesoderm, but only in a region of the ectoderm where Dorsal and Twist are also present

(Fig 1G). Intriguingly, ectopic *sim* expression appeared to decrease after ~30 min of continued photo-activation (Fig 1H and Movie EV2), approximating its wild-type pattern of expression in a single row of cells (Movie EV3). In summary, these results demonstrate that opto-Notch is a suitable tool to study Notch signalling input–output relationships and they suggest transcriptional adaptation to NICD stimulation over time.

### Transcriptional activation in response to NICD nuclear translocation undergoes adaptation

To quantify *sim* expression in response to NICD nuclear translocation, we developed an image analysis pipeline (see Materials and Methods) which confirmed that *sim* expression started a few minutes after photo-activation, reached a plateau at 10–15 min and then started decreasing at ~20 min to reach its lowest level of expression after ~35 min (Fig 2A). Quantification of NICD localisation demonstrated stable nuclear localisation after 10 min of photo-activation and throughout the remainder of the experiment (Fig 2D, and Movie EV4). This reduction in *sim* expression despite stable NICD nuclear localisation could indicate either that nuclei were not competent to transcribe *sim* after ~20 min because some other required factor was not present anymore at that point, or alternatively that nuclei started to adapt to NICD. Late photo-activation after ~25 min from the beginning of cellularisation demonstrated that nuclei were still competent to transcribe *sim* (Fig 2B, and Movies EV5 and EV6), thus suggesting that continuous photo-activation caused nuclear adaptation to NICD. If this was the case, one prediction is that discontinuous pulsatile light inputs should prevent adaptation as NICD would move out of the nuclei in the dark. To test this hypothesis, we administered 5 min pulses of photo-activation followed by a 10 min interval of darkness (Fig 2C and Movies EV7 and EV8). This protocol was established by considering several factors including the overall time window for performing this experiment before the onset of gastrulation (~40 min), the kinetics of *sim* expression (Fig 2A), and the levels of NICD in the nucleus, which drop substantially after 10 min in the dark (Fig EV1A). Pulsatile optogenetic activation resulted in NICD cycling in and out of the nuclei (Fig 2E and Movie EV9) and in stable, non-adaptive *sim* expression (Fig 2C), supporting the conclusion that NICD nuclear accumulation causes transcriptional adaptation over time.

### Mathematical modelling reveals that NICD adaptation fits to known circuits responsible for generating near-perfect adaptation

These results reveal an unexpected dynamic input–output relationship linking Notch signalling and *sim* expression, suggesting that a linear pathway which directly transfers information from the plasma membrane to the nucleus can implement adaptation through transcriptional regulation. To elucidate the regulatory principles that mediate adaptation to NICD, we used mathematical modelling to fit the kinetics of *sim* transcription to motifs known to implement near-perfect or perfect adaptation. Although there are many potential circuits that can generate such adaptation, prior theoretical and experimental work has indicated that they can generally be reduced to three main elementary motifs: the negative feedback loop,

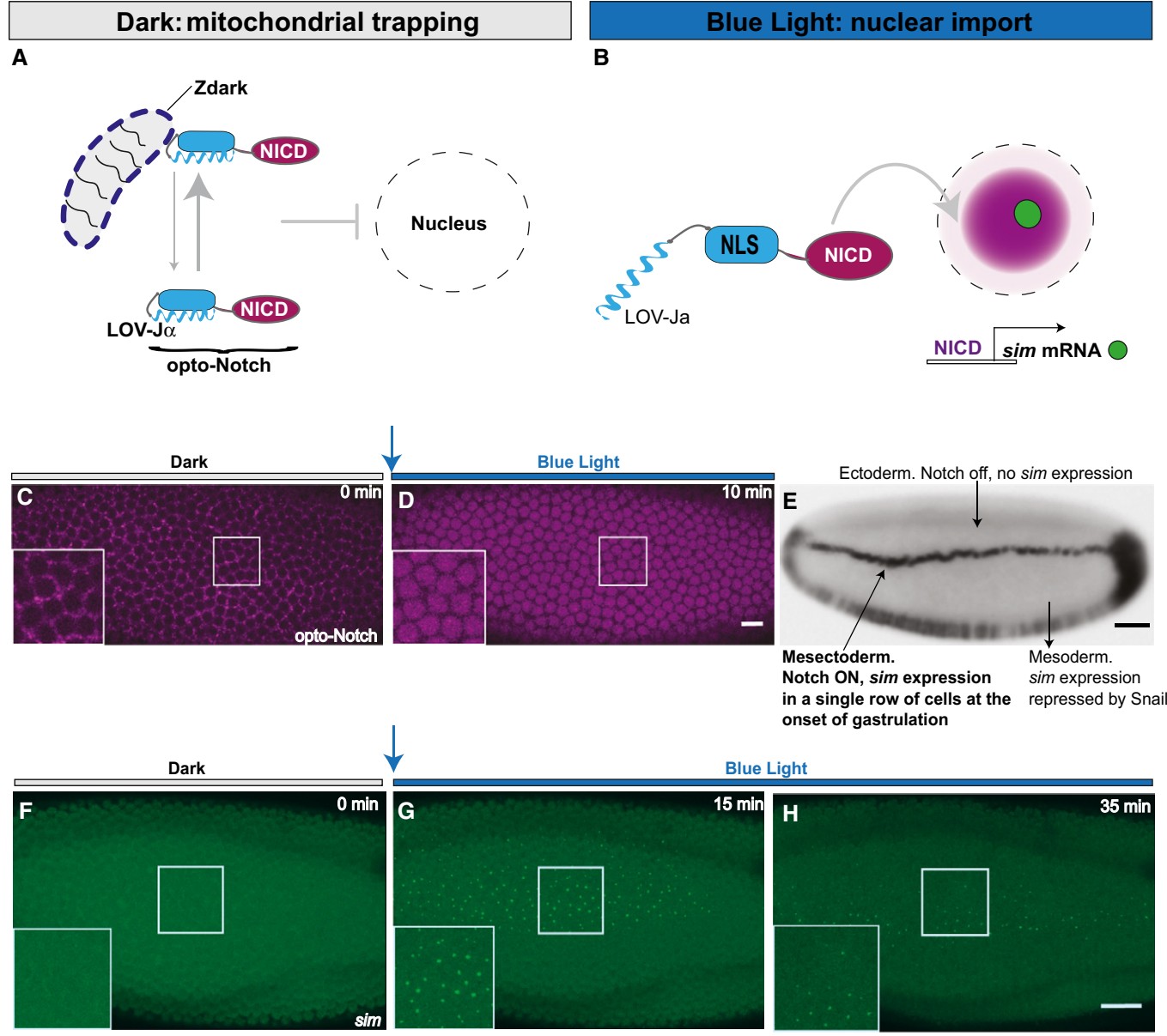

**Figure 1. Optogenetic induction of NICD nuclear translocation and target gene expression with minute-scale precision in live *Drosophila* embryos.**

A, B  Illustration depicting the optogenetic system employed to control NICD nuclear translocation (opto-Notch). NICD (purple) with its endogenous NLS deleted was tagged with the AsLOV2 domain (turquoise) from *Avena Sativa* in such a way that its Jα helix masks an engineered NLS in the dark (A). To ensure maximal nuclear exclusion, Zdark (blue dashed line on the mitochondrion), a protein domain which binds specifically to the dark conformation of AsLOV2, was localised to mitochondria through a TOM20 targeting motif (not depicted). Blue light illumination triggers a conformational change in the LOV2 domain, which results in the simultaneous unbinding from mitochondria and exposure of the engineered NLS, thus allowing NICD nuclear translocation and target gene (*sim*) expression, appearing as a green dot in the nucleus (B).

C, D  Snapshots from confocal live imaging movies of a representative embryo (N = 7) during cellularisation co-expressing opto-Notch, Mito-Zdk, MCP::GFP and sim-MS2. Continuous photo-activation from the onset of cellularisation with a stack size of $z = 25$ μm ($\lambda = 488$ nm) was alternated with mCherry excitation ($\lambda = 561$ nm) at 1 min intervals. Single confocal z-slices of the mCherry channel are shown to record opto-Notch localisation before, and 10 min after sustained photo-activation. Magnified insets show opto-Notch bound to the mitochondrial anchor in the dark (C) and in the nucleus upon photo-activation (D). Scale bar, 10 μm.

E  Wild-type sim expression profile in a *Drosophila* embryo at the onset of gastrulation as revealed by in situ hybridisation. Scale bar, 50 μm.

F–H  Maximum intensity z-projections ($z = 25$ μm) of the same embryo as in (C,D) simultaneously photoactivated and imaged in the GFP channel ($\lambda = 488$ nm) to record *sim* expression at a time resolution of 30 s. Shown are the first cycle of photo-activation, which approximates the dark state (F), 15 min (G) and 35 min (H) after continuous photo-activation. Magnified insets show that sim is not expressed in the dark at this stage (F), but it is expressed in lateral ectodermal cells after 15 min of photo-activation (G) and appeared to be shut off in most nuclei after 35 min (H). Scale bar, 10 μm.

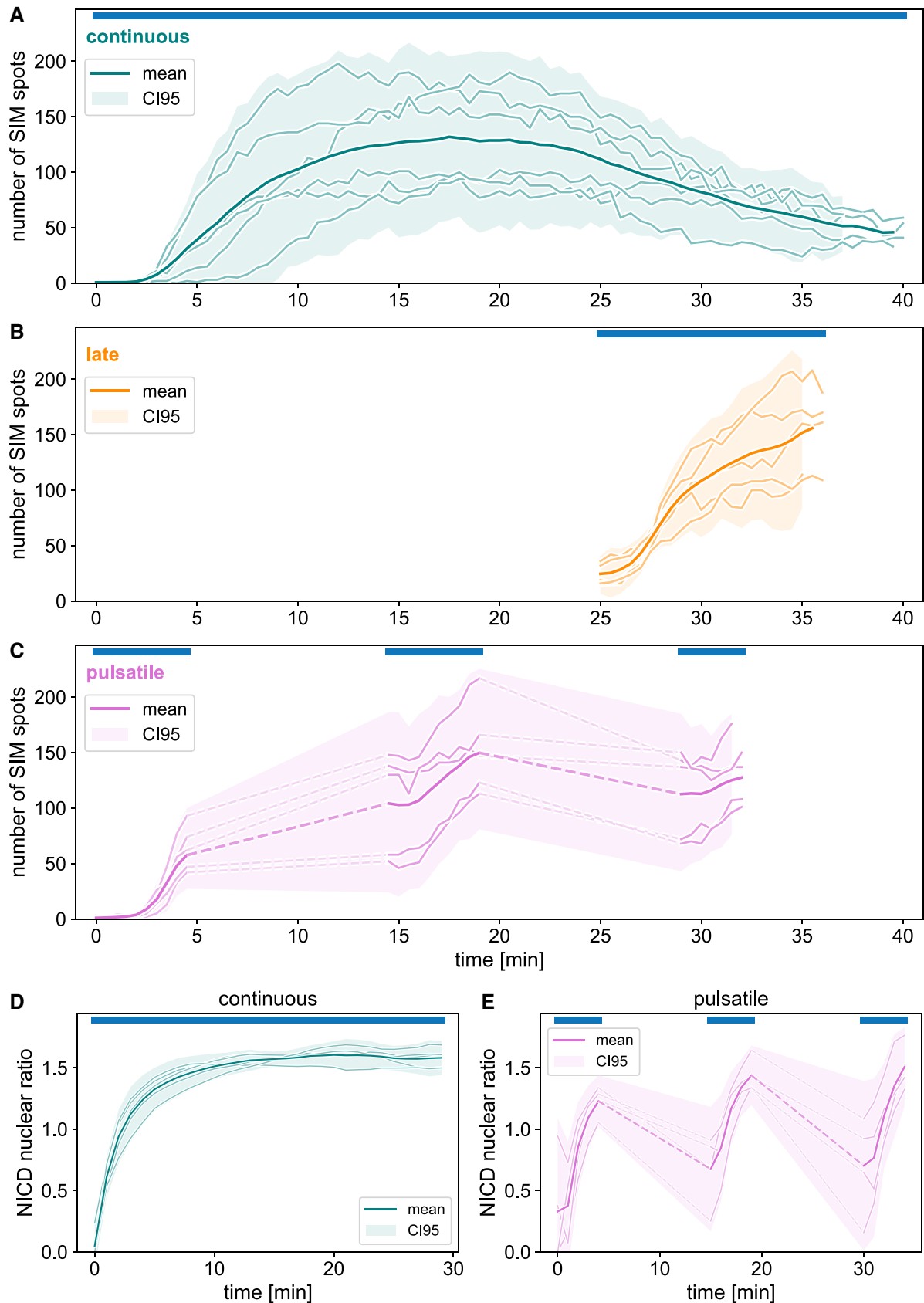

**Figure 2.**

**Figure 2.   Transcriptional activation in response to NICD nuclear translocation undergoes near-perfect adaptation.**

A–C   The number of *sim*-expressing nuclei as counted by automated spot detection is shown over time for different optogenetic activation regimes (top blue bars). (A) Continuous opto-Notch activation initially induces *sim* expression across a large number of nuclei. Over time, the majority of nuclei adapt to the input and terminate expression. (B) Inactivation is not due to developmental timing, as nuclei remain responsive when activation is delayed to a later time point. (C) Sustained nuclear localisation of NICD is required for adaptation to take place, as pulsatile activation can overcome adaptation. Note that *sim* expression could not be measured during dark phases of pulsatile photo-activation as its visualisation requires 488 λ nm excitation which causes photo-activation; dashed lines are a visual aid connecting data from the same embryo.

D, E   Quantification of nuclear localisation of mCherry::NICD during opto-Notch photo-activation. Raw data were bleaching-corrected with an exponential fit and shown is the nuclear/total ratio, which further normalises bleaching and embryo-to-embryo variability. NICD displays persistent nuclear enrichment during continuous activation (D) and shuttles back and forth between nucleus and cytoplasm under the pulsatile protocol (E).

incoherent feedforward regulation and state-dependent inactivation (Barkai & Leibler, 1997; Mangan & Alon, 2003; Friedlander & Brenner, 2009; Ma *et al*, 2009; Ferrell, 2016) We began our analysis with the most parsimonious version of each of these motifs (Fig 3A–I) and hence did not consider more sophisticated configurations such as Antithetical Integral Control (Aoki *et al*, 2019). In negative feedback, an input activates a component (A) which in turn activates both the output and a negative regulator (B) of its own activity (Fig 3A). In feedforward regulation, an input activates two components (A) and (B), of which (A) activates the output and (B) inhibits (A) (Fig 3B). In state-dependent inactivation, an input activates a component (A) that controls the output and can switch into an inhibited (IN) state without further signalling input. The IN state prevents renewed activation for the duration of some refractory period, after which (A) may return to the OFF state (Fig 3C).

We began by simulating all three of these motifs using simple ordinary differential equations (ODEs) based on elementary mass action kinetics and performed automated parameter searches to look for regimes that fit the observed adaptive dynamics of *sim* expression. At this stage, we did not assign specific molecular identities to components (A) and (B) to avoid bias. In accordance with the relatively short time scales involved, we chose equations that best reflect an implementation of the motifs by protein–protein interactions rather than by gene regulatory networks. We considered nuclear NICD levels as the input to the motif and component (A) as a direct regulator of promoter activity, meaning the level of (A) is expected to be proportional to the rate of *sim* mRNA production as measured by the MS2-MCP system. To get an accurate time profile of nuclear NICD levels (i.e. the model input) under continuous and pulsatile optogenetic activation, we performed separate experiments to quantify import and export rates and built a simple

2-compartment model capturing NICD import–export dynamics (Fig EV1A and B, and Movies EV4 and EV9) (see Materials and Methods for more details). Since all nuclei in a sample receive a similar dose of NICD activation, we use bulk *sim* expression as the output measurement for model fitting, as this averages out input-independent fluctuations exhibited by individual cells due to transcriptional bursting (Falo-Sanjuan *et al*, 2019). Our model thus captures the average input–output mapping across a population of activated nuclei.

We found that all three motifs were capable of reproducing *sim* expression dynamics under continuous optogenetic activation and—without further parameter optimisation—also qualitatively reproduced the response to pulsatile activation (Fig 3D–I), albeit at a slightly lower level than the average of experimentally measured time profiles. We also tested the effect of allowing more complex interactions, i.e. Hill coefficients greater or smaller than 1.0. Although the inclusion of these additional degrees of freedom enabled slightly better fits under continuous activation, the predictions for *sim* expression under pulsatile input were further reduced (Fig EV2A–F). Taken together, these results indicate that all three basic adaptation motifs behave similarly and can in principle explain transcriptional adaptation to NICD.

### NICD adaptation is best explained by state-dependent inactivation

To narrow down the most likely candidate motif at play during Notch signalling, we tested whether the three motifs would respond differently to various simulated perturbations (Fig 4A–C). Specifically, we checked the effects of removing negative regulators of *sim* expression, including the feedback and feedforward inhibitors (B)

**Figure 3.   Modelling shows that target gene transcriptional dynamics fit motifs known to underlie near-perfect adaptation.**

A–C   Illustrations of the three simple and fundamental motifs that frequently reoccur in perfect or near-perfect adaptation (modified from Ferrell, 2016): negative feedback (A), incoherent feedforward (B) and state-dependent inactivation (C). Here, the input is the concentration of nuclear NICD and the output is target gene promoter activity, i.e. the rate of *sim* mRNA production (not to be confused with total levels of *sim* mRNA in the cell). The components (A) and (B) in these motifs are placeholders for single or compound molecular regulators; at this point, no specific assumptions are made in our models about their identity. Note that in (C), $A_{in}$ stands for the inhibited state of (A), which can no longer be activated. The dashed line back to $A_{off}$ indicates recovery at slow time scales relative to the motif's adaptation dynamics.

D–I   Results of elementary mass action ODE model fitting to *sim* adaptation dynamics. The input (NICD nuclear concentration; dashed blue line) is modelled based on separate experiments elucidating opto-Notch import–export dynamics (see Fig EV1 and Materials and Methods). *Sim* expression is total MCP::GFP spot intensity, linearly rescaled to match the non-dimensional scale of the ODE models (see Material and Methods for details). (D, F, H) Despite their simplicity, the models of all three motifs can readily be fitted to *sim* adaptation dynamics. (E, G, I) Without further fitting, the resulting models also approximate the loss of adaptation during pulsatile activation, although the model predictions are slightly lower and slightly less responsive to the second and third pulse than experimentally measured nuclei, which may be due to the simplifying assumptions made in the equations describing the motifs.

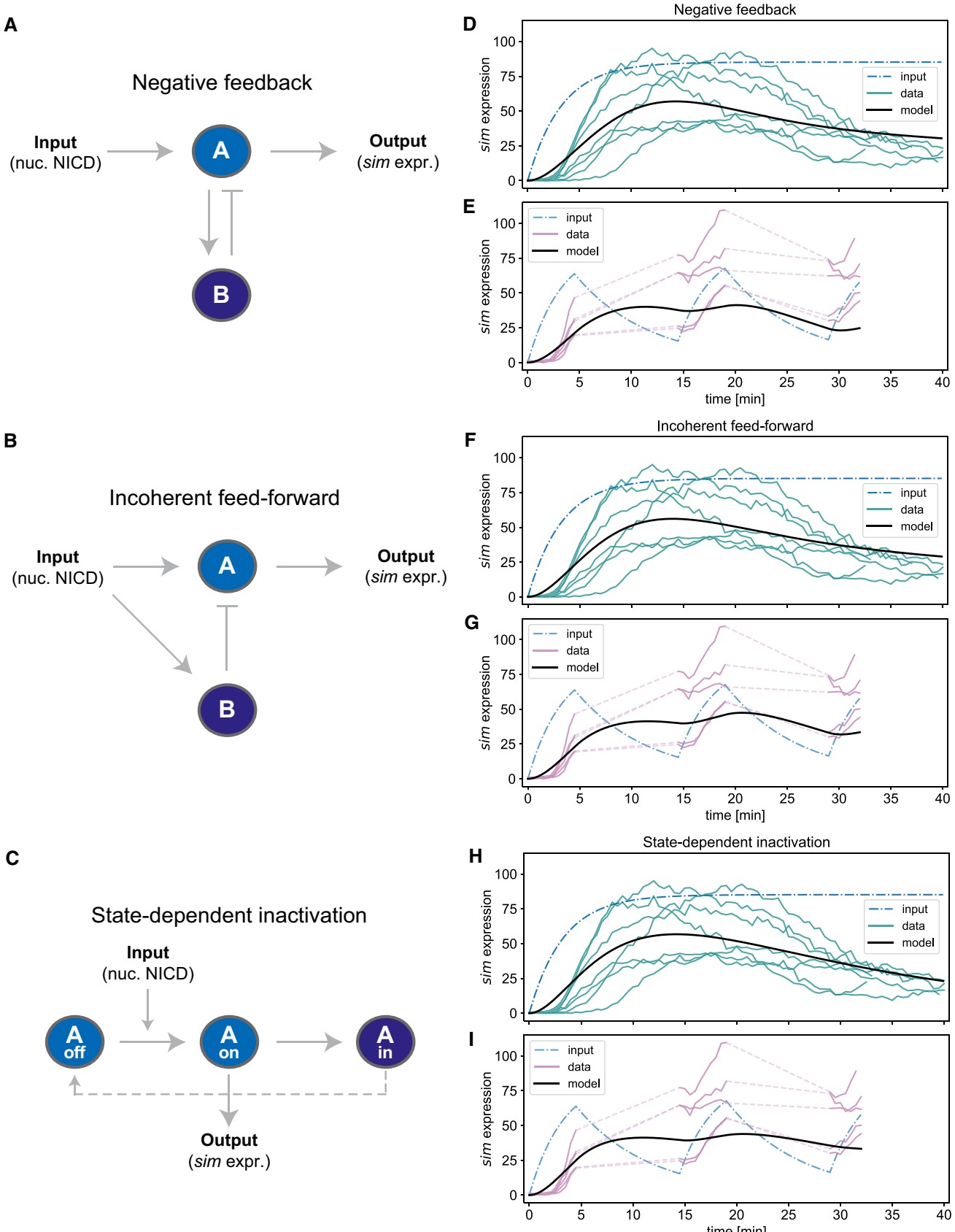

Figure 3.

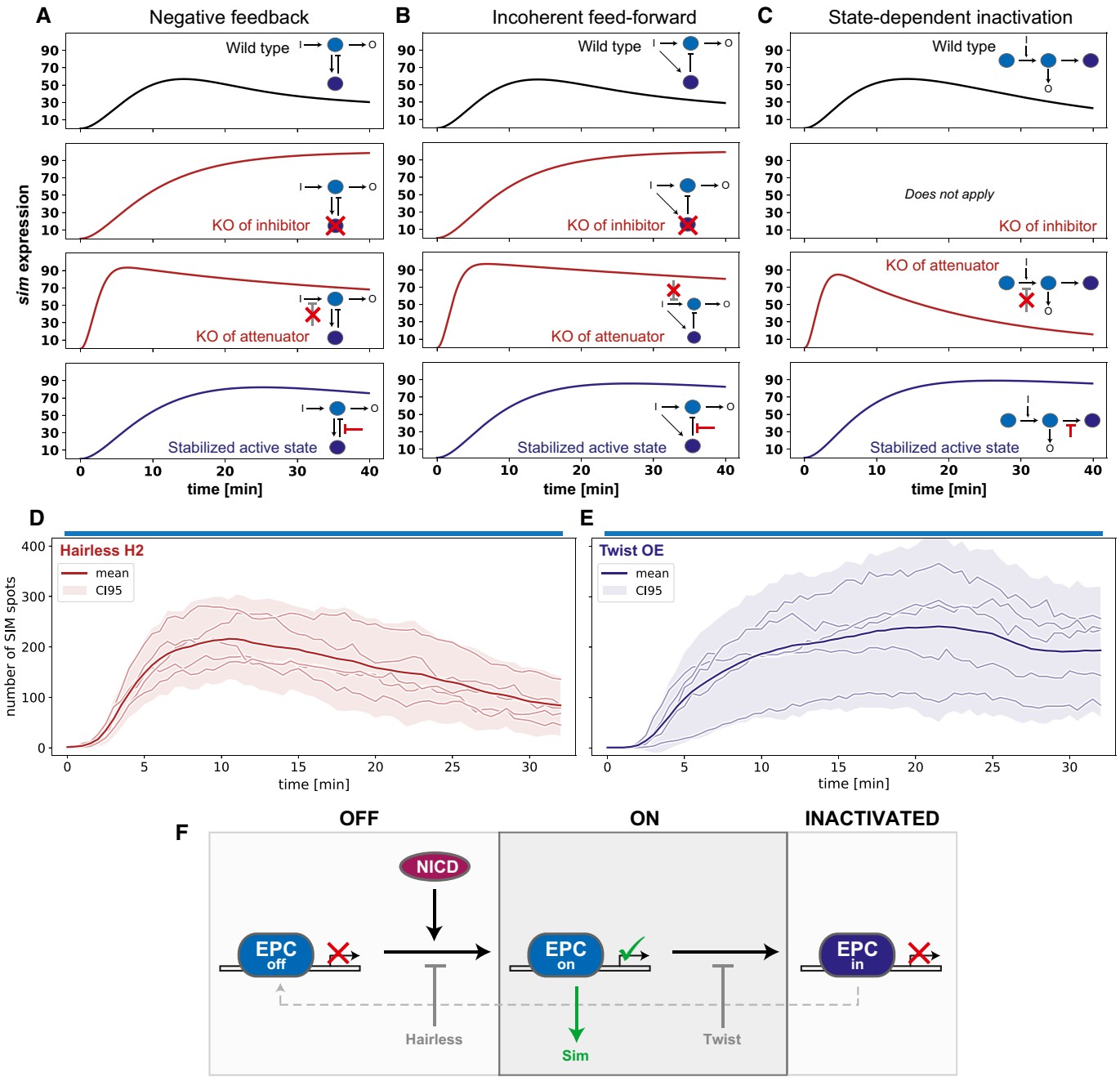

**Figure 4. Molecular perturbations matched to model predictions reveal that adaptation is best explained by state-dependent inactivation.**

A–C   Simulated predictions of sim expression dynamics during continuous opto-Notch activation under different molecular perturbations. Row 1 shows the wild-type model behaviours. Rows 2–3 show the response to a knock-out of a negative regulator of *sim* expression, which could be either the internal inhibitor (B) of the adaptation motifs (row 2) or an attenuator that in wild-type dampens the activation of (A) (row 3). Row 4 simulates the stabilisation of (A) in its active state by reduction of either the inhibitory action of (B) or the auto-inactivation of (A). Note that these perturbations generally lead to a loss of adaptive behaviour, with the exception of the attenuator knock-out on state-dependent inactivation (C; row 3).

D, E   Results of continuous activation experiments in a Hairless heterozygous hypomorphic mutant (Hairless H2) and under overexpression of Twist (Twist OE), which could act as a stabiliser of (A). The observed outcomes are most consistent with the predictions from the state-dependent inactivation model.

F   Proposed model of Notch nuclear adaptation: NICD induces target gene expression by triggering an active state of the enhancer-promoter complex (EPC). Continuous activation leads to a state-dependent transition of the EPC into an inactivated refractory state. Hairless reduces the effective activation rate whereas Twist reduces the effective inactivation rate.

and a putative attenuator of the NICD input on (A). We also simulated the effect of increasing the stability of component (A), the direct controller of *sim* expression. Removal of the negative regulator (B) resulted in a loss of adaptation for the negative feedback and feedforward motifs (Fig 4A and B) and could not be tested for the state-dependent inactivation motif, as in this case (A) itself transits into an inhibited state. Removal of a putative attenuator also resulted in a lack of adaptation for the negative feedback and feedforward motifs, but caused faster activation and higher output levels without preventing adaptation for the state-dependent inactivation motif (Fig 4C). All motifs responded similarly to the stabilisation of (A), namely by overcoming adaptation and increasing output levels compared to wild-type (Fig 4A–C). Equivalent simulations on the more complex Hill models produced similar results: a lack of adaptation when the inhibitor (B) was removed, while removal of the attenuator caused a substantial reduction in adaptation only in the incoherent feedforward motif (Fig EV3A–C). Stabilisation of component (A) reduced adaptation for all three motifs, but unlike in the elementary mass action models, output levels increased only for state-dependent inactivation (Fig EV3A–C). Thus, our model predicts that it is difficult to distinguish the feedback and feedforward motifs through perturbations, whereas state-dependent inactivation can be clearly separated from the two, primarily by the persistence of adaptation when negative regulators of *sim* are removed, as well as by an increased output when (A) is stabilised.

To test these predictions experimentally, we combined molecular perturbations with optogenetic control of Notch input. A well-established negative regulator of Notch signalling is Hairless (H), which competes with NICD to bind Su(H) and inhibits transcription of target genes (Barolo *et al*, 2002). In principle, Hairless could take the role of the inhibitor (B) in the negative feedback and feedforward motifs, as there is evidence that NICD increases, directly or indirectly, Hairless binding to Notch target genes (Gomez-Lamarca *et al*, 2018). Alternatively, Hairless might act as an attenuator on the input in any of the three motifs. The transcription factor Twist is a suitable molecular tool to test the consequences of stabilisation of (A), as mutations in its binding sites in the *sim* enhancer cause a loss of wild-type *sim* expression in the mesectoderm, yet on its own Twist is not capable of activating *sim* expression (Kasai *et al*, 1998; Falo-Sanjuan *et al*, 2019). We generated opto-Notch embryos either heterozygous for a hypomorphic allele of Hairless (H[2]) (Bang & Posakony, 1992) or overexpressing Twist under Gal4-UAS and measured *sim* expression by live imaging upon optogenetic activation. Downregulation of Hairless activity did not alter *sim* expression in the dark (Fig EV4A and B), but caused a faster activation rate (59.4 ± 12.2 nuclei/min compared to 28.3 ± 8.6 nuclei/min in WT; all numbers are mean±st.dev.) and an increase of ˜1.5 fold in the number of nuclei actively transcribing *sim* in response to light-induced NICD nuclear translocation, when compared to wild-type opto-Notch embryos (Fig 4D and Movie EV10). Crucially, Hairless downregulation did not overcome adaptation (Fig 4D). These results were further confirmed by testing an additional allele of Hairless (H[LD]) carrying a point mutation which abolishes binding to Su(H) (Praxenthaler *et al*, 2015) (Fig EV5). Conversely, Twist overexpression overcame nuclear adaptation and caused a ˜ 2-fold increase in peak mean *sim* expression in response to optogenetic activation (Fig 4E, Movie EV11, and Fig EV4C) when compared to control embryos (Fig 2A). Both Hairless downregulation and Twist

overexpression also appeared to cause an expansion of *sim* expression into a more dorsal region of the ectoderm.

Among the motifs we considered, the results of these combined molecular and optogenetic perturbations are most consistent with the predictions from the state-dependent inactivation model. In the elementary mass action models, the only motif that increased *sim* expression but retained adaptation upon removal of the attenuator is the state-dependent inactivation motif (Fig 4C). In the Hill models, both negative feedback and state-dependent inactivation still exhibited substantial adaptation when the attenuator was removed. However, stabilisation of the active state caused an increase in output levels only in the state-dependent inactivation regime, which matches what we observed experimentally upon Twist overexpression (Fig EV3C). Thus, we propose that the *sim cis*-regulatory *region in complex with Su(H) and presumably other trans-acting factors undergoes state-dependent inactivation upon continuous NICD stimulation, switching first from an OFF to an ON state (in which sim* is transcribed) and then transitioning into an inhibited refractory state (Fig 4F). Transition into the inactive state can be prevented by tissue-specific transcription factors such as Twist and more generally adaptation can be ameliorated if NICD input is discontinuous, preventing the IN state from saturating and ultimately allowing the system time to cycle back to the OFF state.

# Discussion

Taken together, the results presented in this study led us to propose a model in which NICD nuclear accumulation triggers state-dependent inactivation of *sim* transcription. This interpretation is supported by the responses to NICD optogenetic stimulation upon Hairless downregulation and Twist overexpression, which are consistent with the predictions of both elementary mass action and Hill-type simulations of the state-dependent inactivation motif. However, it must be kept in mind that our ODE model is a coarse-grained approximation of the true dynamics occurring in the nucleus and specifically at the enhancer–promoter complex. Although this approximation captures the overall adaptation dynamics and responses to perturbations, it does not do so perfectly. In particular, the elementary mass action version does not fully recapitulate the dynamics at the beginning of activation (Fig 3 H) and both model versions—especially the Hill-type version—fall short of producing the experimentally observed levels of *sim* expression under pulsatile activation (Figs 3I and EV2F). It is likely that some mechanism not included in our model, such as local transcription priming (Falo-Sanjuan *et al*, 2019), further modulates state-dependent inactivation. Indeed, the true regulatory motif might be far more complex, with multiple additional components producing an input–output relationship that—although matching the predictions of state-dependent inactivation within the scope of the experiments we performed—in truth does not reduce to one of the three motifs tested here. Along similar lines, we cannot rule out that downregulation of as yet unknown negative regulators other than Hairless might yield an outcome favouring other motifs over state-dependent inactivation. However, Hairless is a well-established negative regulator of Notch signalling (Bang & Posakony, 1992), and our results are fully consistent with a previous report demonstrating its role in repressing *sim* expression in the early embryo

(Morel *et al*, 2001). All of this considered, state-dependent inactivation is at the present stage of our knowledge the most parsimonious model explaining how NICD nuclear adaptation is implemented at the level of *sim* transcriptional regulation. Future studies should seek to elucidate the exact molecular mechanism, which will resolve the open questions discussed above.

The adaptive response induced by continuous NICD stimulation ultimately results in one or two rows of cells expressing *sim*, partially resembling its normal pattern of expression (Fig 1E and H, Movie EV2). In wild-type embryos, the transcription factor Snail represses *sim* expression in the mesoderm (ventral region of the embryo) and cell non-autonomously activates Notch signalling in the mesectoderm (Cowden & Levine, 2002; Bardin & Schweisguth, 2006; De Renzis *et al*, 2006). There, NICD activates *sim* transcription in concert with Dorsal and Twist (Kasai *et al*, 1998). Twist expression is restricted to the mesoderm and a few additional rows of cells immediately dorsally (Kosman *et al*, 1991), which explains why the stripe that resists adaptation under continuous activation is slightly wider than the single row of *sim* expression observed in wild-type embryos, consistent with the role of Twist in our model as a stabiliser of *sim* expression. The transcription factor Dorsal forms a gradient along the entire dorso-ventral axis of the embryo, with higher concentrations in ventral nuclei. Dorsal levels decrease over time in dorso-lateral regions (Reeves, Trisnadi *et al*, 2012), so a simple alternative explanation for the reduction in *sim* transcription during continuous NICD stimulation would be that over time Dorsal concentration drops below the levels needed to support *sim* transcription. However, this explanation is inconsistent with the demonstration that late and pulsatile optogenetic activation induce *sim* transcription in nuclei which would otherwise undergo adaptation (Fig 2B and C). Furthermore, both Twist overexpression and Hairless downregulation cause an expansion of *sim* expression more dorsally compared to NICD optogenetic stimulation alone. These two outcomes are likely due to the overexpression of the stabiliser Twist in cells that do not normally express it and due to the reduced level of Hairless activity which has been suggested to render the *sim cis*-regulatory *region more sensitive to Dorsal (Ozdemir et al*, 2014), respectively.

Dynamic nuclear adaptation might explain how Notch signalling, a relatively simple pathway operating through the cleaved cytoplasmic tail of a surface receptor which directly controls target gene expression (Bray & Furriols, 2001), can dynamically self-terminate or persist in a context-dependent fashion. On the one hand, the presence of factors such as Twist allows the continuous expression of target genes by counteracting adaptation. On the other hand, adaptation could translate a constant Notch signalling input into a pulsatile output by defining a refractory period during which cells are insensitive to the input. This might be particularly relevant during oscillatory developmental processes such as somitogenesis in vertebrates, which require Notch activity for the cyclic generation of new somites (Liao *et al*, 2016; Sonnen *et al*, 2018). In addition, discontinuous optogenetic activation also shows that pulsatile NICD inputs can counteract adaptation, demonstrating yet another layer of dynamic regulation through which the output of Notch signalling can be modified. While adaptation is an important property of many signalling systems, so is the ability to overcome it when necessary. For example, migrating cells moving along a chemokine gradient regulate receptor internalisation and recycling back to the plasma membrane to overcome signalling adaptation and move over long distances (Wong *et al*, 2020). Here our results suggest that a linear signal transduction system, which does not involve signal amplification or other forms of relay through cytoplasmic proteins, can implement dynamic adaptation in the nucleus at the most downstream level of developmental signalling regulation.

# Materials and Methods

### Live imaging and optogenetics

All imaging and optogenetic experiments were done as previously described (Viswanathan *et al*, 2019). Briefly, Cages with flies expressing the opto-Notch module, the *sim* MS2 reporter and MCP:: GFP were kept in the dark in a box. Cycle 14 embryos were selected under halocarbon oil and mounted using a stereomicroscope equipped with a red-emitting LED as the transmission light source (Guglielmi & De Renzis, 2017). Embryos were dechorionated with 100% sodium hypochlorite for 2 min, rinsed with water and mounted immersed in PBS onto a 35 mm glass-bottom dish (MatTek corporation). Embryos were then positioned with their ventro-lateral side facing the cover-slip. Photo-activation and acquisition of movies were done at 22°C with a Zeiss LSM 780 confocal microscope (Carl Zeiss) using a 40×/NA 1.1 water immersion objective (Carl Zeiss). Bright field illumination was filtered using a Deep Amber lighting filter (Cabledelight, Ltd.) in order to locate the embryos. The Microscope was controlled using Zen Black software (Carl Zeiss), and photo-activation protocols were carried out with the Pipeline Constructor Macro (Politi *et al*, 2018). For all experiments, an initial mCherry frame was acquired ($\lambda = 561$ nm) prior to photo-activation. Photo-activation was carried out by illumination with a 488 nm laser line (pixel dwell time = 0.79 microsec, laser power was set at 10% corresponding to an out of the objective power of 35.6 $\mu$W) with a stack size of z = 25 $\mu$m from the apical surface. Photo-activation was alternated with mCherry excitation ($\lambda = 561$ nm) to record NICD localisation at 1 min intervals as a single confocal z-slice. *sim* nuclear spots were simultaneously recorded also using a 488 nm laser at a time resolution of 30 s and a stack size of 25 $\mu$m (z-interval of 1 $\mu$m). Imaging and photo-activation were started either at the onset of cycle 14 (early) or when the cellularisation furrow reached the base of the nuclei (late) and photo-activation was continuous until the onset of ventral furrow formation. For pulsatile activation, each pulse lasted for a period of 5 min following which the embryo was kept in the dark for 10 min prior to the next activation round. For measuring NICD import and export kinetics, the same photo-activation protocol alternated with mCherry excitation (as mentioned in all the experiments above) was followed for a period of 10 min. After either 5 min or 10 in the dark, a final acquisition of both the His::GFP signal ($\lambda = 488$ nm) and mCherry signal ($\lambda = 561$ nm) was recorded.

### Image analysis and modelling

#### Research software stack

Image analysis, data analysis, numerical simulations and data visualisation were all performed using the Anaconda distribution (Anaconda, Inc., Austin, US-TX) of python 3.7 (64-bit) (Python Software

Foundation, Beaverton, US-OR). The following scientific libraries and modules were used: numpy 1.16.4 (Harris *et al*, 2020) for numerical computation, scikit-image 0.13.0 (van der Walt *et al*, 2014) for image processing, sympy 1.4 for symbolic mathematics (Meurer *et al*, 2017), several scipy 1.2.1 modules (Virtanen *et al*, 2020) for image processing, numerical solving of differential equations and parameter fitting, and matplotlib 3.1.0 for plotting. Jupyter Notebooks (jupyter 1.0.0, notebook 6.0.0) were utilised to develop and run all code. Git 2.24.1.windows.2 (github.com/git/git) was employed for version control.

### Sim MS2-MCP spot detection and quantification

We implemented a spot detection pipeline to identify, count and quantify *sim* expression. Image stacks were maximum z-projected and adaptive background subtraction was performed by smoothing the image with a Gaussian filter with σ = 10 and then subtracting this strongly blurred image from the original. Spots were then detected using the scikit-image *blob_log* spot detector. This pipeline was applied independently to every time point in each time course. The number of spots detected at each time point was counted and the median intensity of each spot was measured within a bounding box of edge length 3σ, where σ is an indicator of spot size returned by *blob_log*. The median intensity was chosen rather than the total or mean intensity to reduce the influence of bounding box size and of peripheral pixels in the bounding box, though the three measures do not behave in qualitatively different ways. Where possible we report spot counts as our primary measure of global *sim* expression, as it is largely independent of fluorescence intensity and thus slightly more robust.2

### Quantification and modelling of NICD nuclear translocation

To quantify mCherry::NICD nuclear translocation during continuous and pulsatile opto-Notch activation experiments (Movies EV4 and EV9), nuclei were segmented based on the Histone::GFP channel by applying Gaussian smoothing (σ = 2) and subsequent thresholding using Otsu's method (Otsu, 1979). mCherry::NICD intensity was measured within segmented nuclei and in the surrounding cytoplasmic space, which was masked by dilating the binary nuclear mask with a disk-shaped structural element (radius = 10px). The lowest mean nuclear intensity per track was considered a proxy for background intensity and subtracted from all measurements of that track. Bleaching was captured by fitting an exponential decay function $I(t) = I_0 \cdot e^{-bt}$ to overall mean intensity ($I$) in the 10–20 min time window of the continuous activation experiment (i.e. in the opto-Notch *on*-state equilibrium). The fitted function was used for an approximate bleaching correction of both continuous and pulsatile mCherry::NICD time courses by adding the predicted bleached fraction of the signal back to the signal itself. Reported in Figs 2D and E, and EV1B is the ratio of nuclear to total signal.

To obtain a quantitative description of NICD nuclear import and export kinetics with and without activating light, a simple two-compartment model was fit to data acquired specifically for this purpose, where 10min of continuous activation was followed by darkness for either 5 or 10 min before a final acquisition (see Fig EV1A). Nuclei were segmented based on Histone::GFP, and mCherry::NICD nuclear and cytoplasmic intensities were measured and background-subtracted as described above. The 2-compartment

model was expressed in the following system of ordinary differential equations (ODEs):

$$\frac{dC_F}{dt} = k_e N_F - k_i C_F - b C_F \qquad (1)$$

$$\frac{dN_F}{dt} = k_i C_F - k_e N_F - b N_F \qquad (2)$$

$$\frac{dC_B}{dt} = k_e N_B - k_i C_B + b C_F \qquad (3)$$

$$\frac{dN_B}{dt} = k_i C_B - k_e N_B + b N_F \qquad (4)$$

where $C$ and $N$ are cytoplasmic and nuclear concentrations, $F$ and $B$ stand for fluorescent (unbleached) and bleached mCherry, $k_i$ and $k_e$ are the rate constants for import and export, and $b$ is the rate of bleaching. This ODE system was integrated automatically using *sympy.dsolve,* and the integrated equations were fit to the data.

Specifically, the model was first fit to the continuous activation data (i.e. the first 10min) to retrieve characteristic rate constants $k_i$ and $k_e$ during light exposure of opto-Notch. For this purpose, the initial values of $N_D$ and $C_D$ were set to zero, whereas those of $N_F$ and $C_F$ were considered free parameters alongside $k_i$ and $k_e$. These free parameters were estimated by minimising the mean squared error between the observed nuclear/total mean intensity ratio and the model's prediction. This ratio was chosen for fitting both because it is non-dimensional and because it normalises for variations in the observed total intensity such as cells moving slightly relative to the plane of image acquisition. Following this first fit, the model was also fitted to export data in order to estimate characteristic rate constants $k_i$ and $k_e$ for opto-Notch in darkness. Specifically, the dataset used for fitting comprised the last time point of the 10 min of continuous activation and end point data from samples that were left either 5 or 10 min in the dark following the initial activation. Initial values for $N_F$, $N_D$, $C_F$ and $C_D$ were taken from the last time point of the *on*-state fit and bleaching factor $b$ was set to zero, leaving only two free parameters. The same ratio-based metric was used for optimisation. The *scipy.optimize.minimize* function was employed for all minimisations, the results of which are shown in Fig EV1A. The fitted parameter values for the import–export ODE model are listed below.

| | Activating Light | | Darkness | |
|---|---|---|---|---|
| | **Value** | **Source** | **Value** | **Source** |
| $k_i$ | 0.2619 | fit to data | 10e-6 | fit to data |
| $k_e$ | 0.0455 | fit to data | 0.1419 | fit to data |
| $N_F$ (t = 0) | 0.0273 | fit to data | 0.2726 | end of activ. light |
| $C_F$ (t = 0) | 0.9112 | fit to data | 0.0623 | end of activ. light |
| $N_D$ (t = 0) | 0.0 | fixed | 0.4902 | end of activ. light |
| $C_D$ (t = 0) | 0.0 | fixed | 0.1122 | end of activ. light |
| b | 0.1029 | fit to continuous activation data | 0.0 | fixed |

 

The three fundamental adaptive motifs (see Fig 3) were modelled using Ordinary Differential Equations (ODEs). We started from a previously published formulation (Ferrell, 2016), which we simplified such that all interactions follow elementary mass action kinetics to study the simplest case possible. To test how greater interaction complexity changes modelling outcomes, we subsequently introduced additional degrees of freedom by allowing Hill exponents in the closed interval $h \in [0.25, 4.0]$ for all interactions except first-order decay terms.

The following ODEs were used for the elementary mass action models:

*Negative feedback motif*

$$\frac{dA}{dt} = k_1 \cdot Input \cdot (1-A) - k_2 \cdot B \cdot A - k_3 \cdot A \tag{5}$$

$$\frac{dB}{dt} = k_4 \cdot A \cdot (1-B) - k_5 \cdot B \tag{6}$$

*Incoherent feedforward motif*

$$\frac{dA}{dt} = k_1 \cdot Input \cdot (1-A) - k_2 \cdot B \cdot A - k_3 \cdot A \tag{7}$$

$$\frac{dB}{dt} = k_4 \cdot Input \cdot (1-B) - k_5 \cdot B \tag{8}$$

*State-dependent inactivation motif*

$$\frac{dA_{on}}{dt} = k_1 \cdot Input \cdot (1-A_{on}-A_{in}) - k_2 \cdot A_{on} - k_3 \cdot A_{on} \tag{9}$$

$$\frac{dA_{in}}{dt} = k_2 \cdot A_{on} - k_4 \cdot A_{in} \tag{10}$$

The following ODEs were used for the more complex Hill models:

*Negative feedback motif*

$$\frac{dA}{dt} = k_1 \cdot Input^{h_1} \cdot (1-A)^{h_2} - k_2 \cdot B^{h_3} \cdot A^{h_4} - k_3 \cdot A \tag{11}$$

$$\frac{dB}{dt} = k_4 \cdot A^{h_5} \cdot (1-B)^{h_6} - k_5 \cdot B \tag{12}$$

*Incoherent feedforward motif*

$$\frac{dA}{dt} = k_1 \cdot Input^{h_1} \cdot (1-A)^{h_2} - k_2 \cdot B^{h_3} \cdot A^{h_4} - k_3 \cdot A \tag{13}$$

$$\frac{dB}{dt} = k_4 \cdot Input^{h_5} \cdot (1-B)^{h_6} - k_5 \cdot B \tag{14}$$

*State-dependent inactivation motif*

$$\frac{dA_{on}}{dt} = k_1 \cdot Input^{h_1} \cdot (1-A_{on}-A_{in})^{h_2} - k_2 \cdot (A_{on})^{h_3} - k_3 \cdot A_{on} \tag{15}$$

$$\frac{dA_{in}}{dt} = k_2 \cdot (A_{on})^{h_3} - k_4 \cdot A_{in} \tag{16}$$

We used *scipy.integrate.odeint* to numerically solve these systems of equations and thereby produce simulations of the time-dependent behaviour of the variables.

The time profiles of the *Input*, which represents nuclear NICD, were determined using the fitted import–export model described in the previous section (see eqns. 1–4), by computing $N(t) = N_F(t) + N_B(t)$. For simulations of continuous optogenetic activation, the fit to the "light" condition was used. For simulations of pulsatile activation, the "light-fitted" and "dark-fitted" import–export equations were used intermittently during/between light pulses, with the end point values of one step being forwarded as initial values to the next. The appropriateness of the resulting *Input* time profiles was confirmed by deriving nuclear/total ratios and comparing them to experimentally measured NICD nuclear/total ratio profiles, finding that they are largely consistent (see Fig EV1B). Residual differences likely result from the approximate nature of bleaching correction on the experimental data as compared to the more precise treatment of bleaching in the import–export model.

### Model fitting and prediction of perturbation outcomes

Models were fitted to total *sim* expression (the sum of spot intensities determined as described above) over time in continuous optogenetic activation experiments. We here used total *sim* expression rather than spot count, as the latter is a discrete measure and thus less suited for ODE fitting. Both spot count and total spot intensity are highly correlated and thus both represent robust measures of *sim* expression. As both data and model output are on arbitrary but different scales, they were linearly rescaled by multiplication with fixed factors (1/300 for data, 100 for model output) to bring them to into a comparable range, which was chosen to be neither very small nor very large to simplify numerical computations. This was done for computational convenience and does not affect model parameters or dynamics. The loss function minimised was the mean squared error between the thus rescaled model output and all data points. Rate constant initial values were based on (Ferrell, 2016), and rate constant ranges were constrained to 6 orders of magnitude (0.0001 to 100). Hill coefficient initial values were set to 1.0, and their range was constrained to 4th-order (0.25 to 4). Minimisation was performed using *scipy.optimize.basinhopping*. The fitted parameter values for ODE models of adaptation are listed below. Note that the degradation of the inhibitors (and thus the recovery of the adaptive system) is being reduced to a minimum, which reflects that our experiments end when near-perfect adaptation is achieved, and hence, recovery must be slow compared to the relatively short time scale of the data.

To simulate dynamics under various perturbations, reaction rates were changed as follows: knock-out of the inhibitor (B) was represented by setting $k_4 = 0.0$ (only in feedback and feedforward), knock-out of a putative attenuator of (A) activation by setting $k_1 = 10 \cdot \hat{k}_1$ and stabilisation of active (A) by setting $k_2 = 0.1 \cdot \hat{k}_2$, where $\hat{k}$ represents the original fitted value.

| | Negative feedback | | Incoherent feedforward | | State-dept. inactivation | |
|---|---|---|---|---|---|---|
| | Elementary | Hill | Elementary | Hill | Elementary | Hill |
| $k_1$ | 0.1358 | 0.6531 | 0.1515 | 0.8325 | 0.1806 | 0.4509 |
| $k_2$ | 99.9944 | 5.6291 | 99.9999 | 0.7501 | 0.0494 | 0.0446 |
| $k_3$ | 0.0001 | 0.1231 | 0.0001 | 0.0496 | 0.0001 | 0.0001 |
| $k_4$ | 0.0002 | 0.0548 | 0.0001 | 0.0857 | 0.0001 | 0.0001 |
| $k_5$ | 0.0001 | 0.0005 | 0.0001 | 0.0001 | – | – |
| $h_1$ | – | 3.5779 | – | 3.9521 | – | 3.4775 |
| $h_2$ | – | 1.4870 | – | 2.3835 | – | 0.9659 |
| $h_3$ | – | 4.0000 | – | 4.0000 | – | 0.7655 |
| $h_4$ | – | 0.2500 | – | 0.6609 | – | – |
| $h_5$ | – | 1.4944 | – | 4.0000 | – | – |
| $h_6$ | – | 0.2500 | – | 0.3056 | – | – |

## Statistics and data visualisation details

Means in Figs 2 and 4D and E were computed as a running mean with a time window of 1min. 95% confidence intervals were determined using *scipy.stats.norm.interval* at each time point. Means are only shown if ≥ 9 data points were present within the 1min window and confidence intervals are only shown if ≥ 5 samples were present at a given time point. Activation rates were determined as the slope of the activation curve at its steepest point early on during activation (between 0 and 7 min), as calculated by *numpy.gradient*.

## Cloning

Cloning procedures were performed using standard molecular biology techniques. For the generation of the opto-Notch construct (NES::mCherry::NICD(mut-nls)::LANS4), the NES (Nuclear Export Sequence) GIDLSGLTLQ from *Drosophila melanogaster* Smad4 was used and cloned upstream of mCherry; the NICD sequence corresponded to aa 1,790–2,703 of the *Drosophila melanogaster* Notch. The two endogenous NLS (Nuclear Localisation Sequence) of NICD (Lieber, Kidd *et al*, 1993) were mutated as follow:

NLS1 (aa 1,832–1,835) from KRQR to AAAA
NLS2 (aa 2,202–2,205) from KKAK to AAAA

The LANS4 (LOV2 domain and engineered NLS) was the same as in (Guntas *et al*, 2015) corresponding to the Addgene plasmid pTriEx-mCherry::LANS4, 60785. The mitochondrial anchor NTOM20::Zdk2 was constructed using the TOM20 N-terminus AA 1–36; sequence from *Drosophila melanogaster* (uniProt: Q95RF6). The Zdk2 sequence: addgene plasmid no. 81011. All constructs cloned into pUASp vector for transgenesis.

## Fly strains and genetics

For Notch activation experiments: early, late and pulsatile:

P[w+, UASp> NES::mCherry::NICD(mut-nls)::LANS4]/ w[*];
P[w+, UASp>NTOM20::Zdk2]/ P[ w+, nosP> MCP-no nls::GFP];

P[w+, mat.tubulin>Gal4::VP16]/+ females were crossed to w;+; Sim-MS2/ males.

For Notch import and export kinetics:

P[w+, UASp> NES::mCherry::NICD(mut-nls)::LANS4]/ w[*];
P[w+, UASp>NTOM20::Zdk2]/ +; P[w+, mat.tubulin>Gal4::VP16]/;
P[w+, H2AvGFP] females were crossed to w1118 males.

For *sim* expression without opto-Notch:

w; P[ w+, nosP> MCP-no nls::GFP]/+; P[w+, mat.tubulin>Gal4:: VP16]/+ females were crossed to w;+; Sim-MS2 males.

Hairless mutant experiment:

P[w+, UASp> NES::mCherry::NICD(mut-nls)::LANS4]/ w[*]; P[w+, UASp>TOM20::Zdk2]/ P[ w+, nosP> MCP-no nls::GFP]; P[w+, mat.tubulin>Gal4::VP16]/ $H^2$ or $H^{LD}$ females were crossed to w;+; Sim-MS2; males.

Twist overexpression experiment:

P[w+, UASp> NES::mCherry::NICD(mut-nls)::LANS4]/ w[*]; P[w+, UASp>TOM20::Zdk2]/ P[ w+, nosP> MCP-no nls::GFP]; P[w+, mat.tubulin>Gal4::VP16]/+ females were crossed to w; UAS-twi; Sim-MS2 males.

Fly stocks
Transgenic flies were generated using standard P-element transformation.

P[w+, UASp> NES::mCherry::NICD(mut-nls)::LANS4] on X chromosome.

w; P[w+, UASp>NTOM20::Zdk2]; +.
w;+;Sim-MS2.
yw; P[ w+, nosP> MCP-no nls::GFP]; +/ +.
w; P[w+,mat.tubulin>Gal4::VP16]; P[w+,mat.tubulin>Gal4::VP16].
$H^2$/ TM6B and $H^{LD}$/ TM6B (Praxenthaler *et al*, 2015).

w*; P[w+, H2AvGFP] corresponds to the Histone::GFP marker (His::GFP).

UAS-Twist on 2nd chromosome.

## Data and code availability

All code and extracted numerical data are available freely under the MIT open source license on GitHub at github.com/WhoIsJack/opto-notch-adaptation.

**Expanded View** for this article is available online.

## Acknowledgements

We thank A. Erzberger, D. Krueger and A. Runge for critical reading of the manuscript and all our colleagues in the EMBL Developmental Biology Unit for helpful discussion. We thank the EMBL Advanced Light Microscopy Facility for their helpful support. This work was supported by EMBL internal funding available to S.D.R. We thank the Bloomington *Drosophila* Stock Center and the *Drosophila* Genomics Resource Center for providing fly stocks and cDNAs. We thank D. Maier for providing the H² and H^LD fly stocks. Open Access funding enabled and organized by Projekt DEAL.

## Author contributions

The experiments were conceived and designed by RV, JH, CPC and SDR. RV performed all the experiments and collected the data with the help of CPC. JH developed the image analysis pipeline for *sim* and NICD dynamics quantifications and implemented the mathematical modelling presented in Figs 3 and 4 with input from SDR. JH, SDR and RV wrote the manuscript.

## Conflict of interest

The authors declare that they have no conflict of interest.

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
