## [Review Process File · The EMBO Journal]

Desensitisation of Notch signalling through dynamic adaptation in the nucleus

Stefano De Renzis, Ranjith Viswanathan, Jonas Hartmann, and Cristina Cartes

DOI: [10.15252/embj.2020107245](https://doi.org/10.15252/embj.2020107245)

Corresponding authors: Stefano De Renzis (derenzis@embl.de) , Jonas Hartmann (jonas.m.hartmann@protonmail.com)

Review Timeline:

Submission Date:	10th Nov 20
Editorial Decision:	17th Dec 20
Revision Received:	3rd May 21
Editorial Decision:	4th Jun 21
Revision Received:	14th Jun 21
Editorial Decision:	21st Jul 21
Revision Received:	21st Jul 21
Accepted:	24th Jul 21

Editor: Ieva Gailite / David del Alamo

Transaction Report:

Dear Stefano,

Thank you for submitting your manuscript for consideration by the EMBO Journal. We have now received comments from two reviewers, which are included below for your information.

Based on the interest expressed in the reviewers' comments and the revision outline you provided during the pre-decision discussion, I would like to invite you to address the issues raised by both reviewers in a revised version of the manuscript.

We have extended our 'scooping protection policy' beyond the usual 3-month revision timeline to cover the period required for a full revision to address the essential experimental issues. This means that competing manuscripts published during revision period will not negatively impact on our assessment of the conceptual advance presented by your study. Please contact me if you see a paper with related content published elsewhere to discuss the appropriate course of action.

When preparing your letter of response to the referees' comments, please bear in mind that this will form part of the Review Process File and will therefore be available online to the community. For more details on our Transparent Editorial Process, please visit our website:

<https://www.embopress.org/page/journal/14602075/authorguide#transparentprocess>

Please feel free to contact me if have any further questions regarding the revision. Thank you for the opportunity to consider your work for publication. I look forward to receiving your revised manuscript.

With best regards,

leva

leva Gailite, PhD
Scientific Editor
The EMBO Journal
Meyerhofstrasse 1
D-69117 Heidelberg
Tel: +4962218891309
i.gailite@embojournal.org

When submitting your revised manuscript, please carefully review the instructions below and include the following items:

- 1) a .docx formatted version of the manuscript text (including legends for main figures, EV figures and tables). Please make sure that the changes are highlighted to be clearly visible.
- 2) individual production quality figure files as .eps, .tif, .jpg (one file per figure).
- 3) a .docx formatted letter INCLUDING the reviewers' reports and your detailed point-by-point

response to their comments. As part of the EMBO Press transparent editorial process, the point-by-point response is part of the Review Process File (RPF), which will be published alongside your paper.

4) a complete author checklist, which you can download from our author guidelines ([https://wol-prod-cdn.literatumonline.com/pb-assets/embo-site/Author Checklist%20-%20EMBO%20J-1561436015657.xlsx](https://wol-prod-cdn.literatumonline.com/pb-assets/embo-site/Author%20Checklist%20-%20EMBO%20J-1561436015657.xlsx)). Please insert information in the checklist that is also reflected in the manuscript. The completed author checklist will also be part of the RPF.

6) Before submitting your revision, primary datasets produced in this study need to be deposited in an appropriate public database (see <https://www.embopress.org/page/journal/14602075/authorguide#datadeposition>). Please remember to provide a reviewer password if the datasets are not yet public. Please note that the Data Availability Section is restricted to new primary data that are part of this study. If no data deposition in external databases is needed for this paper, please then state in this section: This study includes no data deposited in external repositories.

7) Our journal encourages inclusion of *data citations in the reference list* to directly cite datasets that were re-used and obtained from public databases. Data citations in the article text are distinct from normal bibliographical citations and should directly link to the database records from which the data can be accessed. In the main text, data citations are formatted as follows: "Data ref: Smith et al, 2001" or "Data ref: NCBI Sequence Read Archive PRJNA342805, 2017". In the Reference list, data citations must be labeled with "[DATASET]". A data reference must provide the database name, accession number/identifiers and a resolvable link to the landing page from which the data can be accessed at the end of the reference. Further instructions are available at .

8) We would also encourage you to include the source data for figure panels that show essential data. Numerical data can be provided as individual .xls or .csv files (including a tab describing the data). For 'blots' or microscopy, uncropped images should be submitted (using a zip archive or a single pdf per main figure if multiple images need to be supplied for one panel). Additional information on source data and instruction on how to label the files are available at .

9) We replaced Supplementary Information with Expanded View (EV) Figures and Tables that are collapsible/expandable online (see examples in <https://www.embopress.org/doi/10.15252/embj.201695874>). A maximum of 5 EV Figures can be typeset. EV Figures should be cited as 'Figure EV1, Figure EV2' etc. in the text and their respective legends should be included in the main text after the legends of regular figures.

10) When assembling figures, please refer to our figure preparation guideline in order to ensure proper formatting and readability in print as well as on screen:
<http://bit.ly/EMBOPressFigurePreparationGuideline>

Please click on the link below to submit the revision online. The revision submission deadline is currently set to 17th Mar 2021. Please contact us if you would need an extension.

Link Not Available

Referee #1:

Viswanathan et al. developed an optogenetic approach for controlling Notch activation by tethering the Notch intracellular domain (NICD) to the mitochondria and sequestering the NLS sequence in the dark state. The authors then quantified sim expression and observed that the system undergoes adaptation when exposed continuously. To determine how this adaptation might occur, the authors used modeling to fit their data to three modeling motifs (circuits) known to generate adaptation: negative feedback, incoherent feedforward, and state-dependent inactivation. The authors claim that the adaptation occurs through state-dependent inactivation based on observations from two perturbations, the reduction of the co-repressor Hairless through introduction of a hypomorph (heterozygous mutant), and the over-expression of the activator Twist. The generation of the optogenetic NICD provides exciting opportunities for future studies, and the paper provides a model for how adaptation of Notch signaling may occur, however there are several deficiencies which detract from the study. In particular, the authors need to (i) expand the analysis to examine how spatial outputs relate to the "adaptation"; (ii) provide more controls for the optogenetic approach; (iii) more clearly explain the modeling (and limitations); and (iv) provide additional evidence that Su(H)'s function as repressor is responsible since Hairless has not been demonstrated to support this role in the embryo (to date).

Major concerns

1. Attention must be paid to the change in spatial outputs of sim expression. The pattern supported by the sim-MS2 reporter starts off broad and then refines into a narrow stripe that abuts the presumptive snail domain in ventral regions. There is little to no discussion of how adaptation really relates to a dorsal repression of the early sim pattern. Furthermore, sim requires Dorsal, and Dorsal levels decrease in lateral regions of the embryo (within the presumptive neurogenic ectoderm) over time (see Reeves et al., 2012; Carrell et al., 2017). Could this "adaptation" relate to

the decrease in Dorsal within this domain that occurs over time? Also, Twist is not expressed in a broad domain (e.g. Fig. 1g) but likely only contributes in the domain supported late (e.g. Fig. 1h). The study would benefit by discussing how adaptation relates to change in space, possibly due to interactions with Dorsal; previous studies have demonstrated that Dorsal and Su(H) (repressor form) can compete to affect the dorsal boundaries of gene along the DV axis including sim (Ozdemir et al., 2014).

In addition, the movies of sim-MS2 in response to pulsed activation of NICD support reporter expression in a broad domain (i.e. early sim response) for early and later pulses, and this result lends further support to the view that the narrowing of the stripe relates to an adaptation response (absent from pulsed input outputs when applied early or late). Pointing this out, and including a discussion of the spatial outputs, would help the authors' argument.

2. The authors show adaptation occurs in figure 2A, when embryos are exposed to blue light for >30 min. There is no mention of a control showing that this level and duration of blue light is not toxic to the embryo, or does not cause photobleaching. These experiments need to be performed in order to rule out that the loss of sim expression occurs because of phototoxicity or photobleaching. The Twi overexpression looks promising, but even it appears to exhibit decreases after ~20-25min.

3. In Figure 1f (which is missing the f label), the authors say that sim expression was measured before photoactivation, effectively in the dark. However, the methods do not detail how this was accomplished. In the methods they state that both photoactivation and capturing sim expression use the 488 nm laser. Thus, it is impossible to capture sim expression in the dark, since the laser used to excite the GFP will also activate the optogenetic tags. The methods need to be updated with clarification on how this experiment was performed or the figure updated to reflect that this wasn't in the dark. Repeating the experiment with a MCP::RFP could be a good control.

4. It is not clear in the main text, figures, or methods section what each species in the models are, making it hard to evaluate whether the models are reasonable. Since the input, A, B, and the output represent actual biological molecules (proteins or mRNA) it should be clearer how the authors interpreted them. While it is stated that the input is nuclear NICD and it is assumed the output is sim expression, it is unclear what A and B are. The models presented here are derived from Michaelis-Menten enzyme kinetics. It is unclear in this paper, but in reference 5 (Ferrell), the output is A. In this case, that means that sim is A. sim is a mRNA molecule and not an enzyme, so modeling this behavior using Michaelis-Menten kinetics seems erroneous or at least an over-simplification. A more rigorous treatment of the problem is necessary to show that these assumptions are reasonable. It is possible that a more complex circuit may simplify to similar expressions, but it is necessary to demonstrate this explicitly.

Along these lines, if Twist is "A" and Su.H-Hairless complex is "B", there is no evidence in the literature to support the idea that B represses A.

5. In Figure 3d-i, the models seem to fit the average experimental data well, but in the pulsatile experiments, the model appears generally lower than the average data. In addition, the levels of the pulsatile model are generally similar to or lower than the adaptation levels of the continuous model. The authors state this is due to simplification or how the input was approximated, however this suggests that none of the models capture the behavior of the real system, and call into question whether the conclusion that NICD adaptation occurs via state-dependent inactivation is accurate. A more complex model, see point 4 above, might better capture these dynamics, and could lend further support for this conclusion.

6. In figure 4c and f the model includes repressive species, which are not present in figure 3c. Are

these species part of the model in figure 3c or only in figure 4c and f? This needs to be clarified, and potentially justified.

7. Both the negative feedback and incoherent feed-forward loop models require Hairless to be activated by NICD or Sim (although the models are vague, see point 2 above), however no evidence is provided to support this assumption. It is possible that removing Hairless in the biological context is not the same as removing B in the simulations. It then becomes difficult to conclude what effect removing Hairless has on the models. Clarification of the model is necessary to further evaluate these experiments.

8. Furthermore, a role for Hairless acting in the early embryo has not been demonstrated (or at least the primary literature was not cited). The data in Fig 4A make a good argument for Hairless acting in the early embryo. However, a second Hairless allele should be used to make sure the phenotype does not relate to a 2nd site mutation; alternatively, Su(H) mutant alleles that do not bind Hairless can be examined (see Praxenthaler et al., 2017).

9. More information about the vantage point in movies is necessary. For example, in the movie in which Twist is overexpressed, does sim-MS2 extend to dorsal regions? Is the movie a projection of a small 25 um section? Similarly, in the Hairless het background, does sim-MS2 extend to the dorsal regions?

10. Line 175. In the Hairless het background, it is stated that the response is faster and of higher output without affecting adaptation. By what measure can you say that adaptation is not affected?

Minor concerns

1. There is a mix of -ize and -ise in the paper. For example, line 80 localization, and line 87 localisation. There might be other instances as well. The authors should be consistent in their chosen spellings.

2. In figure 2d, e, and f, the baseline NICD nuclear ratio is around 50%. However, during the dark phases of the pulsatile experiment (f), the ratio drops below 25%. What is the explanation for this?

3. The NICD nuclear ratio should be calculated over time and plotted in a similar manner to sim. From movie 1 it appears that the levels increase and then decrease. Could this change in NICD nuclear levels contribute to the sim expression dynamics? In other words, sim dynamics reflects nuclear NICD dynamics and possibly that of other inputs.

Referee #2:

Referee Report for Desensitisation of Notch signalling through dynamic adaptation in the nucleus

In this work, Viswanathan et al. apply optogenetics to understand how the Notch signaling pathway is read-out. This is an important question, given the broad role that Notch plays in development. The approaches are generally performed to a high standard and the work has potential to be impactful. The combination of experiment and theory is good and adds value to the work.

I do have a number of concerns - particularly with regard to Fig. 3 and the lack of dynamics under pulsatile activation - that should be addressed to improve the manuscript.

Major comments

1. The import/export rates need to be better quantified. Line 90 "it rapidly (<2min) translocated" - but I cannot see the quantitative evidence for this (Fig. 1d shows too coarse data). The intensity inside the nucleus should be quantified and then the time course of importation should be analyzed to deduce a better estimate of the import/export rates. This is important for understanding the later modeling.
2. I find Fig 2c confusing. Why is only data shown during illumination? Then, in Fig 2f there's the nuclear ratio also in the "dark". Can the full time courses be shown in Fig 2c? Further, why was the particular pulsing protocol used? This depends on the import/export rates, but, as stated above, those need improved quantification (just showing a single video is not sufficient).
3. The input-output relation used in Fig. 3 is too simplistic. Such import/export profiles are typically exponential-like. The use of linear is an unnecessary simplification as the input/output rates can be measured directly (point 1 above).
4. My main point of disagreement with the authors is their statement "Our model thus captures ..." (line 162). The fits in Fig. 3e,g,i are far from capturing the data. They seem like the dynamics in the model are too slow - which is maybe what's needed to get the long tail in Fig. 3d,f,h. Overall, I find the fitting in Fig. 3 not especially convincing.
5. Related to the above, the data in Fig. 3e,g,i appears slightly odd. Particularly for the middle peak (Time 15-20 mins) it looks like 4 distinct sets of data points - I assume this corresponds to 4 different embryos. The sample variability appears to be very large. A more detailed analysis of variability is required so the reader can gauge the reproducibility of the results. Further, it may make sense to normalize the data in a manner to make the model fitting more robust.
6. In the image analysis protocol (line 275-278), the signal itself is used to segment nuclei (if I'm understanding correctly). Better would be to cross in a far-red histone marker and then use that to reliably segment nuclei. This can also act to correct for image intensity variations. If I'm understanding correctly, this is also why Fig. 2c only has disjointed data shown. However, the absence of such a time course makes it difficult to draw conclusions about the system dynamics.
7. Given the rapid dynamics and switch-like behavior, it seems odd to me that the authors used simple Michaelis-Menten-like kinetics (e.g. line 294). Why not explore non-linear (Hill function) feedback? This seems more realistic. More detailed model analysis is required to substantiate the conclusion that state-dependent inactivation is the most likely model - with the current evidence, this conclusion is not fully supported.

Minor comments

1. Given that there are not severe referencing limits, the authors should cite relevant literature more completely. For example, in the introduction reviews (13-15) are cited, not primary literature. In particular, there has been substantial previous work using optogenetics to understand gene regulation in vivo and these should be properly included (e.g. Sako et al. Cell Reports 2016, Huang et al. ELife 2017, McDaniel et al. Mol. Cell. 2019, Johnson et al. Current Biology 2020). Another example is line 152, where Sorre et al. Dev Cell 2014 should be cited who introduced this model as a means to interpret morphogen rate of change.
2. Line 84 - "trough" is typo
3. Line 88 - "particulate" should be "particular"
4. Ordering of Figure 1 - "d" comes before "c" in text, which is odd and makes reading confusing.
5. In the text, where something is described in Methods add a pointer. For example, line 117 an image analysis pipeline is referred to. It turns out to be in the Methods but not clear from existing text.

6. Line 317, unclear to me what the rescaling is and why. More explanation needed.
7. Ref 5 is incomplete.
8. This may be simply due to the conversion process, but the image quality of the graphs is poor in the pdf.

Referee #1:

Viswanathan et al. developed an optogenetic approach for controlling Notch activation by tethering the Notch intracellular domain (NICD) to the mitochondria and sequestering the NLS sequence in the dark state. The authors then quantified sim expression and observed that the system undergoes adaptation when exposed continuously. To determine how this adaptation might occur, the authors used modeling to fit their data to three modeling motifs (circuits) known to generate adaptation: negative feedback, incoherent feedforward, and state-dependent inactivation. The authors claim that the adaptation occurs through state-dependent inactivation based on observations from two perturbations, the reduction of the co-repressor Hairless through introduction of a hypomorph (heterozygous mutant), and the over-expression of the activator Twist. The generation of the optogenetic NICD provides exciting opportunities for future studies, and the paper provides a model for how adaptation of Notch signaling may occur, however there are several deficiencies which detract from the study. In particular, the authors need to (i) expand the analysis to examine how spatial outputs relate to the "adaptation"; (ii) provide more controls for the optogenetic approach; (iii) more clearly explain the modeling (and limitations); and (iv) provide additional evidence that Su(H)'s function as repressor is responsible since Hairless has not been demonstrated to support this role in the embryo (to date).

We thank this reviewer for their suggestions. We have taken into account all the comments that have been raised. We performed new experiments to support the role of Hairless and to obtain more precise measurements of nuclear import/export kinetics. This allowed us to improve the modelling results, which we also further expanded upon with simulations based on Hill kinetics. Finally, we discussed in detail any potential limitations, as requested.

Major concerns

1. Attention must be paid to the change in spatial outputs of sim expression. The pattern supported by the sim-MS2 reporter starts off broad and then refines into a narrow stripe that abuts the presumptive snail domain in ventral regions. There is little to no discussion of how adaptation really relates to a dorsal repression of the early sim pattern. Furthermore, sim requires Dorsal, and Dorsal levels decrease in lateral regions of the embryo (within the presumptive neurogenic ectoderm) over time (see Reeves et al., 2012; Carrell et al., 2017). Could this "adaptation" relate to the decrease in Dorsal within this domain that occurs over time? Also, Twist is not expressed in a broad domain (e.g. Fig. 1g) but likely only contributes in the domain supported late (e.g. Fig. 1h). The study would benefit by discussing how adaptation relates to change in space, possibly due to interactions with Dorsal; previous studies have demonstrated that Dorsal and Su(H) (repressor form) can compete to affect the dorsal boundaries of gene along the DV axis including sim (Ozdemir et al., 2014). In addition, the movies of sim-MS2 in response to pulsed activation of NICD support reporter expression in a broad domain (i.e. early sim response) for early and later pulses, and this result lends further support to the view that the narrowing of the stripe relates to an adaptation response (absent from pulsed input outputs when

applied early or late). Pointing this out, and including a discussion of the spatial outputs, would help the authors' argument.

We agree with this comment and now discuss this aspect both in the results and in the discussion, where we elaborate on how adaptation relates to dorso-ventral patterning of *sim* expression, Twist and Dorsal. We cited the relevant papers as suggested.

“The adaptive response induced by continuous NICD stimulation ultimately results in one or two rows of cells expressing *sim*, partially resembling its normal pattern of expression (**Fig. 1E,H and Movie EV2**). In wild type embryos, the transcription factor Snail represses *sim* expression in the mesoderm (ventral region of the embryo) and cell non-autonomously activates Notch signalling in the mesectoderm (Bardin & Schweisguth, 2006, Cowden & Levine, 2002, De Renzis et al., 2006). There, NICD activates *sim* transcription in concert with Dorsal and Twist (Kasai et al., 1998). Twist expression is restricted to the mesoderm and a few additional rows of cells immediately dorsally, which explains why the stripe that resists adaptation under continuous activation is slightly wider than the single row of *sim* expression observed in wild-type embryos, consistent with the role of Twist in our model as a stabiliser of *sim* expression. The transcription factor Dorsal forms a gradient along the entire dorso-ventral axis of the embryo, with higher concentrations in ventral nuclei. Dorsal levels decrease over time in dorso-lateral regions (Reeves, Trisnadi et al., 2012), so a simple alternative explanation for the reduction in *sim* transcription during continuous NICD stimulation would be that over time Dorsal concentration drops below the levels needed to support *sim* transcription. However, this explanation is inconsistent with the demonstration that late and pulsatile optogenetic activation induce *sim* transcription in nuclei which would otherwise undergo adaptation (**Fig. 2B,C**). Furthermore, both Twist overexpression and Hairless downregulation cause an expansion of *sim* expression more dorsally compared to NICD optogenetic stimulation alone. These two outcomes are likely due to the overexpression of the stabiliser Twist in cells that do not normally express it and due to the reduced level of Hairless activity which has been suggested to render the *sim cis*-regulatory region more sensitive to Dorsal (Ozdemir, Ma et al., 2014), respectively.”

2. The authors show adaptation occurs in figure 2A, when embryos are exposed to blue light for >30 min. There is no mention of a control showing that this level and duration of blue light is not toxic to the embryo, or does not cause photobleaching. These experiments need to be performed in order to rule out that the loss of *sim* expression occurs because of phototoxicity or photobleaching. The Twi overexpression looks promising, but even it appears to exhibit decreases after ~20-25min.

The experiments are internally controlled as adaptation occurs only in dorsolateral nuclei while *sim* expression in the mesectoderm is maintained. Furthermore, all embryos gastrulated, indicating that they developed normally (gastrulation is a very sensitive process). Finally, as the reviewer already noted, Twist overexpression indicates that the dynamics of *sim* expression are not dominated by bleaching. As requested, we added Movie EV3 demonstrating wild-type pattern of *sim* expression upon photo-activation (without the opto-Notch system).

3. In Figure 1f (which is missing the f label), the authors say that *sim* expression was measured before photoactivation, effectively in the dark. However, the methods do not detail how this was accomplished. In the methods they state that both photoactivation and capturing *sim* expression use the 488 nm laser. Thus, it is impossible to capture *sim* expression in the dark, since the laser used to excite the GFP will also activate the optogenetic tags. The methods need to be updated with clarification on how this experiment was performed or the figure updated to reflect that this wasn't in the dark. Repeating the experiment with a MCP::RFP could be a good control.

We have clarified this point in the relevant legend (see below). The first snapshot simultaneously acts as the first activation step, so it approximates the dark condition as *sim* expression starts only after several minutes of photo-activation. Indeed, in the first snapshot there is no *sim* expression (Fig. 1F).

“(F-H) Maximum intensity z-projections ($z = 25 \mu\text{m}$) of the same embryo as in (C,D) simultaneously photoactivated and imaged in the GFP channel ($\lambda=488 \text{ nm}$) to record *sim* expression at a time resolution of 30 s. Shown are the first cycle of photo-activation, which approximates the dark state (F), 15 min (G), and 35 min (H) after continuous photo-activation.”

4. It is not clear in the main text, figures, or methods section what each species in the models are, making it hard to evaluate whether the models are reasonable. Since the input, A, B, and the output represent actual biological molecules (proteins or mRNA) it should be clearer how the authors interpreted them. While it is stated that the input is nuclear NICD and it is assumed the output is *sim* expression, it is unclear what A and B are. The models presented here are derived from Michaelis-Menten enzyme kinetics. It is unclear in this paper, but in reference 5 (Ferrell), the output is A. In this case, that means that *sim* is A. *sim* is a mRNA molecule and not an enzyme, so modeling this behavior using Michaelis-Menten kinetics seems erroneous or at least an over-simplification. A more rigorous treatment of the problem is necessary to show that these assumptions are reasonable. It is possible that a more complex circuit may simplify to similar expressions, but it is necessary to demonstrate this explicitly.

Along these lines, if Twist is "A" and Su.H-Hairless complex is "B", there is no evidence in the literature to support the idea that B represses A.

Based on this feedback as well as further suggestions below and by reviewer 2, we made numerous changes to the models and to our way of describing them in the text. We believe that this strengthens the approach, the conclusions and the clarity with which they are presented.

Our initial approach was based on equations described previously in the field, which included Michaelis-Menten terms (Ferrell, 2016, Ma et al., 2009). We now decided to instead simplify these terms, creating the most basic conceivable implementations of the three motifs using elementary mass action kinetics alone. We find that these motifs are still capable of approximating *sim* expression dynamics in much the same

way. We then re-introduced greater interaction complexity through Hill coefficients representing cooperativity, to test whether this changes the outcome (new Fig. EV 2 and EV3). The results remain in support of the conclusion that adaptation is best explained by state-dependent inactivation. To briefly clarify some of the specific points raised by the reviewer:

We did not initially assign any specific molecular identities to components A and B, as making such assumptions was not necessary to build the models. Only in Fig. 4 did we include Hairless and tested whether it might take the role of the inhibitor (B) or of an attenuator of the activation of (A), finding that only the latter is consistent with the results.

We did not suppose that *sim* itself is (A). Instead, we supposed that (A) is the direct regulator of *sim* transcription, meaning it should be proportional to the measured output (i.e. MCP::GFP, a measure of the rate of *sim* transcription).

Similarly, we do not suggest that Twist is (A) and that the Su(H)-Hairless complex might be (B).

We recognise that in the initial manuscript these points were not presented as clearly as we would have liked. Therefore, as mentioned previously, we have made numerous amendments and extensions to better explain our approach, for instance:

Results related to Fig3:

"We began by simulating all three of these motifs using simple Ordinary Differential Equations (ODEs) based on elementary mass action kinetics and performed automated parameter searches to look for regimes that fit the observed adaptive dynamics of *sim* expression. At this stage, we did not assign specific molecular identities to components (A) and (B) to avoid bias. We considered nuclear NICD levels as the input to the motif and component (A) as a direct regulator of promoter activity, meaning the level of (A) is expected to be proportional to the rate of *sim* mRNA production as measured by the MS2-MCP system. To get an accurate time profile of nuclear NICD levels (i.e. the model input) under continuous and pulsatile optogenetic activation, we performed separate experiments to quantify import and export rates and built a simple 2-compartment model capturing NICD import-export dynamics (**Fig. EV1A,B, and Movies EV4,9**) (see Material and Methods for more details). Since all nuclei in a sample receive a similar dose of NICD activation, we use bulk *sim* expression as the output measurement for model fitting, as this averages out input-independent fluctuations exhibited by individual cells due to transcriptional bursting (Falo-Sanjuan et al., 2019). Our model thus captures the average input-output mapping across a population of activated nuclei."

Results related to Fig4:

"To narrow down the most likely candidate motif at play during Notch signalling, we tested whether the three motifs would respond differently to various simulated perturbations (**Fig. 4A-C**). Specifically, we checked the effects of removing negative regulators of *sim* expression, including the feedback and feedforward inhibitors (B) as well as a putative attenuator of the NICD input on (A). We also simulated the effect of increasing the stability of component (A), the direct controller of *sim* expression. Removal of the negative regulator (B) resulted in a loss of adaptation for the negative feedback and feedforward motifs (**Fig. 4A,B**) and could not be tested for the state-dependent inactivation motif, as in this case (A) itself transits into an inhibited state. Removal of a putative attenuator also resulted in a lack of adaptation

for the negative feedback and feedforward motifs, but caused faster activation and higher output levels without preventing adaptation for the state-dependent inactivation motif (**Fig. 4C**). All motifs responded similarly to the stabilisation of (A), namely by overcoming adaptation and increasing output levels compared to wild-type (**Fig. 4A-C**). Equivalent simulations on the more complex Hill models produced similar results: a lack of adaptation when the inhibitor (B) was removed, while removal of the attenuator caused a substantial reduction in adaptation only in the incoherent feedforward motif (**Fig. EV3 A-C**). Stabilisation of component (A) reduced adaptation for all three motifs, but unlike in the elementary mass action models, output levels increased only for state-dependent inactivation (**Fig. EV3A-C**). Thus, our model predicts that it is difficult to distinguish the feedback and feedforward motifs through perturbations, whereas state-dependent inactivation can be clearly separated from the two, primarily by the persistence of adaptation when negative regulators of *sim* are removed, as well as by an increased output when (A) is stabilised.”

5. In Figure 3d-i, the models seem to fit the average experimental data well, but in the pulsatile experiments, the model appears generally lower than the average data. In addition, the levels of the pulsatile model are generally similar to or lower than the adaptation levels of the continuous model. The authors state this is due to simplification or how the input was approximated, however this suggests that none of the models capture the behavior of the real system, and call into question whether the conclusion that NICD adaptation occurs via state-dependent inactivation is accurate. A more complex model, see point 4 above, might better capture these dynamics, and could lend further support for this conclusion.

We performed new experiments to more accurately quantify NICD import/export rates (New Fig. 2D,E and EV1). We used these new data to model a more appropriate input function for our adaptation motif simulations. This resulted in a better fit for both the continuous and pulsatile conditions, although the pulsatile fits were still slightly below the average experimentally measure time profiles. Interestingly, allowing for more interaction complexity through Hill coefficients made this outcome worse, not better (New Fig. EV2).

We now clarify this aspect in the discussion:

“However, it must be kept in mind that our ODE model is a coarse-grained approximation of the true dynamics occurring in the nucleus and specifically at the enhancer-promoter complex. Although this approximation captures the overall adaptation dynamics and responses to perturbations, it does not do so perfectly. In particular, the elementary mass action version does not fully recapitulate the dynamics at the beginning of activation (**Fig. 3H**) and both model versions – especially the Hill-type version – fall short of producing the experimentally observed levels of *sim* expression under pulsatile activation (**Figs. 3I, EV2F**).”

6. In figure 4c and f the model includes repressive species, which are not present in figure 3c. Are these species part of the model in figure 3c or only in figure 4c and f? This needs to be clarified, and potentially justified.

The addition of these repressive species as modulators of the motifs in question is based on the simulated and empirical perturbation experiments we performed in this section. They are not directly part of the models but take the form of alterations in model parameters reflecting the perturbations we performed. More details on this are now given in the methods section:

"To simulate dynamics under various perturbations, reaction rates were changed as follows: knock-out of the inhibitor (B) was represented by setting $k_4 = 0.0$ (only in feedback and feedforward), knock-out of a putative attenuator of (A) activation by setting $k_1 = 10 \cdot \widehat{k}_1$ and stabilisation of active (A) by setting $k_2 = 0.1 \cdot \widehat{k}_2$, where \widehat{k} represents the original fitted value."

7. Both the negative feedback and incoherent feed-forward loop models require Hairless to be activated by NCID or Sim (although the models are vague, see point 2 above), however no evidence is provided to support this assumption. It is possible that removing Hairless in the biological context is not the same as removing B in the simulations. It then becomes difficult to conclude what effect removing Hairless has on the models. Clarification of the model is necessary to further evaluate these experiments.

To address this point, we have now included additional simulations in Fig. 4 which consider Hairless as an attenuator of the input, as explained in the results:

"In principle, Hairless could take the role of the inhibitor (B) in the negative feedback and feedforward motifs, as there is evidence that NICD increases, directly or indirectly, Hairless binding to Notch target genes (Gomez-Lamarca et al., 2018). Alternatively, Hairless might act as an attenuator on the input in any of the three motifs."

We found that the feedback and feedforward circuit do not fit the data even if Hairless fulfils its repressive role as a passive attenuator rather than as the induced inhibitor (B). Thus, we ultimately conclude that Hairless is most likely an attenuator on the input of state-dependent inactivation, consistent with the reviewer's expectations.

We now also mention this point in the discussion:

"Along similar lines, we cannot rule out that downregulation of as yet unknown negative regulators other than Hairless might yield an outcome favouring other motifs over state-dependent inactivation. However, Hairless is a well-established negative regulator of Notch signalling (Bang & Posakony, 1992), and our results are fully consistent with a previous report demonstrating its role in repressing *sim* expression in the early embryo (Morel, Lecourtois et al., 2001)."

8. Furthermore, a role for Hairless acting in the early embryo has not been demonstrated (or at least the primary literature was not cited). The data in Fig 4A make a good argument for Hairless acting in the early embryo. However, a second Hairless allele should be used to make sure the phenotype does not relate to a 2nd

site mutation; alternatively, Su(H) mutant alleles that do not bind Hairless can be examined (see Praxenthaler et al., 2017).

Hairless does act as an inhibitor of *sim* expression in the embryo, as previously reported (Morel, Lecourtois et al., 2001). As requested, we have confirmed our results with a second allele of Hairless that does not bind to Su(H); see new Fig EV5.

9. More information about the vantage point in movies is necessary. For example, in the movie in which Twist is overexpressed, does *sim*-MS2 extend to dorsal regions? Is the movie a projection of a small 25 μ m section? Similarly, in the Hairless het background, does *sim*-MS2 extend to the dorsal regions?

The embryos were mounted with the ventrolateral surface touching the coverslip so that the mesectoderm could be imaged (this information is included in the relevant movie legends). The expression of *sim* appeared to be expanded dorsally upon Twist overexpression or Hairless downregulation, but we did not investigate the details of this expansion as this is beyond the scope of our study. As already clarified in point 1, we commented on this expansion both in the results and in the discussion as much as the focus of this study permits.

10. Line 175. In the Hairless het background, it is stated that the response is faster and of higher output without affecting adaptation. By what measure can you say that adaptation is not affected?

We reword this sentence as: "Hairless downregulation did not overcome adaptation"

Minor concerns

1. There is a mix of -ize and -ise in the paper. For example, line 80 localization, and line 87 localisation. There might be other instances as well. The authors should be consistent in their chosen spellings.

This is now fixed, we only use "ise".

2. In figure 2d, e, and f, the baseline NICD nuclear ratio is around 50%. However, during the dark phases of the pulsatile experiment (f), the ratio drops below 25%. What is the explanation for this?

This panel is no longer displayed. We performed new experiments to quantify nuclear NICD using Histone-GFP as a nuclear marker (Fig. 2D,E and EV1).

3. The NICD nuclear ratio should be calculated over time and plotted in a similar manner to *sim*. From movie 1 it appears that the levels increase and then decrease. Could this change in NICD nuclear levels contribute to the *sim* expression dynamics? In other words, *sim* dynamics reflects nuclear NICD dynamics and possibly that of other inputs.

As suggested we quantified NICD nuclear ratio over time. We used Histone-GFP to segment the nuclei; these new results are reported in Fig. 2D and demonstrate stable NICD nuclear accumulation over time.

Referee #2:

Referee Report for Desensitisation of Notch signalling through dynamic adaptation in the nucleus

In this work, Viswanathan et al. apply optogenetics to understand how the Notch signaling pathway is read-out. This is an important question, given the broad role that Notch plays in development. The approaches are generally performed to a high standard and the work has potential to be impactful. The combination of experiment and theory is good and adds value to the work.

I do have a number of concerns - particularly with regard to Fig. 3 and the lack of dynamics under pulsatile activation - that should be addressed to improve the manuscript.

We performed new experiments to address the concerns related to Fig. 3, in particular we quantified NICD nuclear import/export rates as requested and added new simulations based on Hill kinetics. Overall, these new measurements have strengthened our conclusions.

Major comments

1. The import/export rates need to be better quantified. Line 90 "it rapidly (<2min) translocated" - but I cannot see the quantitative evidence for this (Fig. 1d shows too coarse data). The intensity inside the nucleus should be quantified and then the time course of importation should be analyzed to deduce a better estimate of the import/export rates. This is important for understanding the later modeling.

To address this point, we quantified NICD import/export rates using Histone-GFP to segment the nuclei and a 2-compartment model to derive import and export rates whilst accounting for bleaching. These new results are presented in Fig. 2D,E and EV1. The calculated rates were used to more accurately represent the input during simulations.

2. I find Fig 2c confusing. Why is only data shown during illumination? Then, in Fig 2f there's the nuclear ratio also in the "dark". Can the full time courses be shown in Fig 2c? Further, why was the particular pulsing protocol used? This depends on the import/export rates, but, as stated above, those need improved quantification (just showing a single video is not sufficient).

We cannot image *sim* expression during the dark phase of the pulsatile protocol, as this would cause photo-activation. We clarified this in the relevant legend:

“Note that *sim* expression could not be measured during dark phases of pulsatile photo-activation as its visualization requires 488 λ nm excitation which causes photo-activation; dashed lines are a visual aid connecting data from the same embryo.”

Regarding the pulsatile protocol, we added the following explanation to the results:

“This protocol was established by considering several factors including the overall time window for performing this experiment before the onset of gastrulation (~40 min), the kinetics of *sim* expression (**Fig. 2A**), and the levels of NICD in the nucleus, which drop substantially after 10 min in the dark (**Fig. EV1A**).”

3. The input-output relation used in Fig. 3 is too simplistic. Such import/export profiles are typically exponential-like. The use of linear is an unnecessary simplification as the input/output rates can be measured directly (point 1 above).

As clarified above in point 1, we followed this suggestion. Please see new Fig. 3 and Fig. 2 D,E and EV1 for the quantifications.

4. My main point of disagreement with the authors is their statement "Our model thus captures ..." (line 162). The fits in Fig. 3e,g,i are far from capturing the data. They seem like the dynamics in the model are too slow - which is maybe what's needed to get the long tail in Fig. 3d,f,h. Overall, I find the fitting in Fig. 3 not especially convincing.

Our new measurements of NICD import/export rates have improved the fitting presented in Fig. 3, although they remain at a slightly lower level than the average of experimentally measured time profiles for the pulsatile fits. We comment on this point in the discussion:

“Although this approximation captures the overall adaptation dynamics and responses to perturbations, it does not do so perfectly. In particular, the elementary mass action version does not fully recapitulate the dynamics at the beginning of activation (**Fig. 3H**) and both model versions – especially the Hill-type version – fall short of producing the experimentally observed levels of *sim* expression under pulsatile activation (**Figs. 3I, EV2F**). It is likely that some mechanism not included in our model, such as local transcription priming (Falo-Sanjuan et al., 2019), further modulates state-dependent inactivation.”

5. Related to the above, the data in Fig. 3e,g,i appears slightly odd. Particularly for the middle peak (Time 15-20 mins) it looks like 4 distinct sets of data points - I assume this corresponds to 4 different embryos. The sample variability appears to be very large. A more detailed analysis of variability is required so the reader can gauge the reproducibility of the results. Further, it may make sense to normalize the data in a manner to make the model fitting more robust.

The measurements presented in Fig. 3E,G,I indeed represent 5 distinct embryos. For clarity, we have now joined together data points from the same embryos.

As these experiments are challenging and laborious, measuring a sufficient number of samples to allow an in-depth analysis of their variability is currently not feasible. However, our visualizations clearly show all our data, as well as statistical indicators such as running mean and confidence intervals where useful (Figs. 2, 4D,E, EV5), giving readers sufficient information to gauge reproducibility.

Finally, although we are sceptical about applying non-uniform normalizations when the sources of variability are unknown, we have tested different ways of normalizing the data for model fitting, such as rescaling individual *sim* expression tracks to have matching minima and maxima. However, this does not make an appreciable qualitative difference in the results of our simulations (data not shown), so we decided to stick with raw data.

6. In the image analysis protocol (line 275-278), the signal itself is used to segment nuclei (if I'm understanding correctly). Better would be to cross in a far-red histone marker and then use that to reliably segment nuclei. This can also act to correct for image intensity variations. If I'm understanding correctly, this is also why Fig. 2c only has disjointed data shown. However, the absence of such a time course makes it difficult to draw conclusions about the system dynamics.

We agree that segmenting and quantifying on the same channel is far from ideal, so we removed this analysis and instead repeated these experiments with Histone-GFP to segment the nuclei and quantify NICD translocation dynamics in general and import/export rates in particular (see new Fig. 2D,E and EV1). Performing these measurements using a far-red histone marker would have further complicated an already sophisticated experiment as we do not have a far-red Histone marker and the required laser lines to perform this experiment on the same set-up used for all the other experiments.

7. Given the rapid dynamics and switch-like behavior, it seems odd to me that the authors used simple Michaelis-Menten-like kinetics (e.g. line 294). Why not explore non-linear (Hill function) feedback? This seems more realistic. More detailed model analysis is required to substantiate the conclusion that state-dependent inactivation is the most likely model - with the current evidence, this conclusion is not fully supported.

Our original approach was to use a previously established mathematical formulation of the motifs (Ferrell, 2016). Based on this feedback and additional points raised by reviewer 1, we decided to change our modelling strategy and begin by reporting an even simpler version of the models with elementary mass action kinetics only. This version reproduced the same results as the previously used Michaelis-Menten version.

We then tested what happens when far more complex interaction kinetics are enabled by making the Hill coefficient a free parameter (within bounds). Although the inclusion of these additional degrees of freedom enabled slightly better fits under continuous activation, the predictions for *sim* expression under pulsatile input were

further reduced (**Fig. EV2**). We also used Hill-based models to predict the outcome of the molecular perturbations (**Fig. EV3**). Overall, the results support the conclusion that state-dependent inactivation is the most likely model. We now discuss potential limitations of our conclusions in the discussion:

“This interpretation is supported by the responses to NICD optogenetic stimulation upon Hairless downregulation and Twist overexpression, which are consistent with the predictions of both elementary mass action and Hill-type simulations of the state-dependent inactivation motif. However, it must be kept in mind that our ODE model is a coarse-grained approximation of the true dynamics occurring in the nucleus and specifically at the enhancer-promoter complex. Although this approximation captures the overall adaptation dynamics and responses to perturbations, it does not do so perfectly. In particular, the elementary mass action version does not fully recapitulate the dynamics at the beginning of activation (**Fig. 3H**) and both model versions – especially the Hill-type version – fall short of producing the experimentally observed levels of *sim* expression under pulsatile activation (**Figs. 3I, EV2F**). It is likely that some mechanism not included in our model, such as local transcription priming (Falo-Sanjuan et al., 2019), further modulates state-dependent inactivation. Indeed, the true regulatory motif might be far more complex, with multiple additional components producing an input-output relationship that – although matching the predictions of state-dependent inactivation within the scope of the experiments we performed – in truth does not reduce to one of the three motifs tested here.”

Minor comments

1. Given that there are not severe referencing limits, the authors should cite relevant literature more completely. For example, in the introduction reviews (13-15) are cited, not primary literature. In particular, there has been substantial previous work using optogenetics to understand gene regulation in vivo and these should be properly included (e.g. Sako et al. Cell Reports 2016, Huang et al. ELife 2017, McDaniel et al. Mol. Cell. 2019, Johnson et al. Current Biology 2020). Another example is line 152, where Sorre et al. Dev Cell 2014 should be cited who introduced this model as a means to interpret morphogen rate of change.

We have added the suggested references.

2. Line 84 - "trough" is typo

Fixed. Thank you.

3. Line 88 - "particulate" should be "particular"

We actually meant “particulate” (minute separate particles). This refers to the localisation of NICD in the cytoplasm on mitochondria.

4. Ordering of Figure 1 - "d" comes before "c" in text, which is odd and makes reading confusing.

We changed the order as suggested.

5. In the text, where something is described in Methods add a pointer. For example, line 117 an image analysis pipeline is referred to. It turns out to be in the Methods but not clear from existing text.

We have added pointers to the methods and in some cases elaborated on our approach directly in the text.

6. Line 317, unclear to me what the rescaling is and why. More explanation needed.

The empirical measurements of *sim* expression and the model predictions are on separate arbitrary scales, hence rescaling is performed to simplify parameter screening. Note that the same factor is used for rescaling of all samples, so the dynamics remain unchanged by this transformation. This point is now clarified in the methods and has been updated to reflect a simplification that was made in newer versions of the code (i.e. the use of explicitly specified rescaling factors), which does not alter any of the results.

7. Ref 5 is incomplete.

Fixed. Thank you.

8. This may be simply due to the conversion process, but the image quality of the graphs is poor in the pdf.

Yes, this is due to conversion. The final figures are provided in high resolution.

Dear Stefano,

Thank you for submitting a revised version of your manuscript. I apologise for the delay in the assessment of your manuscript due to delayed submission of referee reports. We have now received input from both original reviewers. As you can see, reviewer #1 finds that several of their concerns have not been sufficiently resolved and still have to be addressed before they can support publication of the manuscript. This opinion was seconded by reviewer #2 in the referee cross-commenting session. I would therefore invite you to address the remaining referee comments and the following editorial issues in the final revision:

- 1) Please submit up to five keywords
- 2) It is mandatory to include a 'Data Availability' section after the Materials and Methods. Before submitting your revision, primary datasets produced in this study need to be deposited in an appropriate public database, and the accession numbers and database listed under 'Data Availability'. Please remember to provide a reviewer password if the datasets are not yet public (see <https://www.embopress.org/page/journal/14602075/authorguide#datadeposition>). In case you have no data that requires deposition in a public database, please state so in this section. Note that the Data Availability Section is restricted to new primary data that are part of this study.
*** Note - All links should resolve to a page where the data can be accessed. ***
- 3) Please zip movie legends together with the corresponding movie files (<https://www.embopress.org/page/journal/14602075/authorguide#expandedview>).
- 4) Papers published in The EMBO Journal are accompanied online by a 'Synopsis' to enhance discoverability of the manuscript. It consists of A) a short (1-2 sentences) summary of the findings and their significance, B) 2-3 bullet points highlighting key results and C) a synopsis image that is 550x300-600 pixels large (width x height, jpeg or png format). You can either show a model or key data in the synopsis image. Please note that the size is rather small and that text needs to be readable at the final size. Please send us this information along with the revised manuscript.

Please feel free to contact me if have any further questions regarding the revision. Thank you again for giving us the chance to consider your manuscript for The EMBO Journal. I am looking forward to receiving the final revised version.

With best regards,

leva

leva Gailite, PhD
Scientific Editor
The EMBO Journal
Meyerhofstrasse 1
D-69117 Heidelberg
Tel: +4962218891309
i.gailite@embojournal.org

Further information is available in our Guide For Authors:

Revision to The EMBO Journal should be submitted online within 90 days, unless an extension has been requested and approved by the editor; please click on the link below to submit the revision online before 2nd Sep 2021:

Link Not Available

Referee #1:

Summary

Viswanathan et al. developed opto-Notch, an optogenetic approach to ectopically activate Notch targets, by tagging the Notch intracellular domain (NICD) with a blue light sensitive tag that prevents nuclear import and sequesters NICD at mitochondria in the dark. They use opto-Notch to investigate expression of a Notch target gene, *sim*. Expression of *sim* was quantified, and adaptation was observed when exposed to continuous NICD. Modeling was used to try and understand how the system might be functioning to achieve adaptation. The authors conclude that adaptation occurs through state-dependent inactivation by using a Hairless hypomorph and Twist over expression. Opto-Notch is an exciting tool that is likely to be used in many future studies on Notch target gene activation. Careful attention to detail is given to the experiments performed, however, the modeling requires additional work. It is appreciated that the authors would like to keep A and B anonymous to avoid bias, however it is unclear what A and B are in a general sense and how they are being treated. This is necessary to evaluate whether the differential equations accurately reflect the system and are being interpreted correctly. For example, is A being converted from an inactive state to an active state upon addition of the input, or is A being produced by the input via transcription and translation. Similarly, the perturbations made to the experimental system are not inherently in their model, so it is hard to conclude that the experimental perturbations match the perturbations in the models. These perturbations are the only support for the authors' conclusion that the state-dependent inactivation model explains *sim* expression, and as such, need to be treated more rigorously.

Major Concerns

1. It seems like the wrong movie references are given and I didn't see any movie legends (which made it more difficult to evaluate the claims, as I had to assume the numbering was off by one). Movies EV1 and EV2 seem to be referenced correctly, but every movie reference after that appears incorrect.
2. It is my understanding that Movie EV3 is a control without opto-Notch. Was this movie quantified? I think quantification and comparison to the opto-Notch could be insightful to understand what is occurring. Similarly, are their controls without opto-Notch for *sim* expression in the Hairless hypomorph and Twist over expression that can be quantified?
3. It appears that there are two types of quantification performed on *sim* expression: number of *sim* spots, and intensity of *sim* spots. It appears that adaptation is occurring in both. One issue is it is

mentioned that a sum of spot intensities is used, but I couldn't find a description in the materials and methods. Also, it is stated that median intensity is used in line 413. Why is the median used instead of the mean? Since it is unclear how levels are quantified, I had a hard time interpreting what adaptation means in both of these contexts. If a nucleus had expression but lost expression, was this considered zero expression and included in the quantification of levels, or were only detectable spots included in the quantification? I think it needs to be clearer how the intensity is quantified since the models do not include a spatial component to them and do not reflect number of sim spots.

4. The Hairless hypomorph and Twi over expression data in figure 4 are presented as number of sim spots. The results from these experiments are used to conclude the state-dependent inactivation model best describes the experimental sim data. However, the models do not contain a spatial component and are actually giving the levels of sim expression, not the number of sim spots. sim expression levels need to be shown in figure 4 instead of number of sim spots.

5. In lines 515-519, I don't understand why the data was rescaled. The model should be fit to the data, so it should match the scale of the data.

6. In general, I have concerns with the implementation of the modeling. I appreciate that A and B are not assigned molecular identities to avoid bias, however I think it needs to be clearly stated whether the models are trying to capture activation of already translated proteins (like a phosphorylation event as a molecular example) or if they are trying to capture activation of genes and subsequent protein translation. Both approaches have their merits, but would require different differential equations and different interpretations.

7. As far as I know, NICD acts as a transcription factor and there is no reason to assume that NICD doesn't act directly on the sim enhancer. Similarly, it is not expected that the NICD would act on other proteins to activate them via something like phosphorylation (however it could form complexes, which would require refining the differential equations). Justification for why it is assumed that NICD acts through an intermediary, A, is necessary, and if A is treated as a transcription factor that is being transcribed due to activation by NICD, the differential equations need to be adjusted.

8. Why are $(1-x)$ terms (lines 480, 483, 486; equations 5, 7, 9) included in the production of A and B? This is not included in the models in figure 3, but implies that additional forms of A and B are considered and this value represents A and B subtracted from a total A or total B which has been normalized to one. This assumption needs to be justified, as I would expect total A and B could be changing at such a dynamic time during embryogenesis. Also, this model strongly suggests A exists in an inactivated form and is being activated by the NICD (as opposed to gene activation, where this model would not be appropriate).

9. Instead of manipulating constants to mimic the effects of an attenuator or stabilizer on A, the attenuator and stabilizer should be included explicitly in the differential equations. The manipulation of constants implies the attenuator and stabilizer are at steady state, which may not be true at such a dynamic time during embryogenesis.

10. It is stated that Twi is necessary for sim expression (line 241), however, in the opto-Notch system, sim is expressed in regions where Twi is not expressed. These statements are contradictory. As far as I know, Twi is a transcription factor, so other than binding in a complex, Twi would have no additional role as a stabilizer. If a complex is being formed, this should be modeled

explicitly. I think this study would benefit from trying to model the activation of sim directly instead of assuming that sim is proportional to A, and to do so by writing their own differential equations instead of using the previously published differential equations that were modified.

Minor Concerns

1. In line 101 and 119 it is not clear in the main text that "dark" refers to the first time point. It would help to add a sentence stating this.
2. Do the authors have an explanation for why ectopic sim expression is lost but the wild type pattern remains? Since adaptation appears to be occurring (at least in the ectopic domain), why doesn't sim turn off in all nuclei instead of restricting to the wild type pattern? I assume the response would be that Twi is stabilizing the system so it cannot adapt. Were the levels in the wild type sim domain looked at by themselves to see if adaptation is occurring there to some degree?
3. At the end of movie EV4 the NICD is no longer enriched in nuclei. Is there an explanation for this?
4. In lines 338-340 the sentence is unclear.
5. Does Twi overexpression turn sna on, and if so, why doesn't this block sim activation?

Referee #2:

Overall, the authors have done a decent job of dealing with the concerns raised. I am still not entirely convinced by the modelling results, but I feel that this can be explored in further work - it is not a reason to oppose publication of the current manuscript.

Heidelberg, 14/06/2021

Reviewer #1**Summary**

Viswanathan et al. developed opto-Notch, an optogenetic approach to ectopically activate Notch targets, by tagging the Notch intracellular domain (NICD) with a blue light sensitive tag that prevents nuclear import and sequesters NICD at mitochondria in the dark. They use opto-Notch to investigate expression of a Notch target gene, *sim*. Expression of *sim* was quantified, and adaptation was observed when exposed to continuous NICD. Modeling was used to try and understand how the system might be functioning to achieve adaptation. The authors conclude that adaptation occurs through state-dependent inactivation by using a Hairless hypomorph and Twist over expression. Opto-Notch is an exciting tool that is likely to be used in many future studies on Notch target gene activation. Careful attention to detail is given to the experiments performed, however, the modeling requires additional work. It is appreciated that the authors would like to keep A and B anonymous to avoid bias, however it is unclear what A and B are in a general sense and how they are being treated. This is necessary to evaluate whether the differential equations accurately reflect the system and are being interpreted correctly. For example, is A being converted from an inactive state to an active state upon addition of the input, or is A being produced by the input via transcription and translation. Similarly, the perturbations made to the experimental system are not inherently in their model, so it is hard to conclude that the experimental perturbations match the perturbations in the models. These perturbations are the only support for the authors' conclusion that the state-dependent inactivation model explains *sim* expression, and as such, need to be treated more rigorously.

Major Concerns

1. It seems like the wrong movie references are given and I didn't see any movie legends (which made it more difficult to evaluate the claims, as I had to assume the numbering was

off by one). Movies EV1 and EV2 seem to be referenced correctly, but every movie reference after that appears incorrect.

We have double-checked this and found that movies and legends are correct. The legends were included at the end of the merged file as requested by the editor. What probably confused the reviewer is that in some but not all movie file names an extra blank space was included, which means that the movies do not get sorted properly in a folder. The movies are however correctly labelled with their EV number. We have now removed the extra blank spaces.

2. It is my understanding that Movie EV3 is a control without opto-Notch. Was this movie quantified? I think quantification and comparison to the opto-Notch could be insightful to understand what is occurring. Similarly, are their controls without opto-Notch for sim expression in the Hairless hypomorph and Twist over expression that can be quantified?

This movie was not quantified as it was acquired to show that our imaging protocol does not cause phototoxicity or alteration of sim expression in a single stripe of cells as requested in the previous revision. We did not acquire movies of Hairless and Twi over-expression without opto-Notch, but only snapshots of embryos (containing opto-Notch and the different alleles but not exposed to light) at the onset of gastrulation to demonstrate that under these conditions sim expression was not expanded (Fig. EV4).

3. It appears that there are two types of quantification performed on sim expression: number of sim spots, and intensity of sim spots. It appears that adaptation is occurring in both. One issue is it is mentioned that a sum of spot intensities is used, but I couldn't find a description in the materials and methods. Also, it is stated that median intensity is used in line 413. Why is the median used instead of the mean? Since it is unclear how levels are quantified, I had a hard time interpreting what adaptation means in both of these contexts. If a nucleus had expression but lost expression, was this considered zero expression and included in the quantification of levels, or were only detectable spots included in the quantification? I think it needs to be clearer how the intensity is quantified since the models do not include a spatial component to them and do not reflect number of sim spots.

Spot count was used where possible because it is the most reliable measure of global activation, seeing that it is independent of fluorescence intensity (except in the rare instances where the intensity is very close to the detection threshold of the spot detection algorithm).

Because spot count is a discrete measure, it was not considered suitable for model fitting. Instead, the sum of spot intensities was used, as it also represents overall expression in the embryo. The mean would not be suitable, as it would increase when a spot is reduced below the spot detection threshold (and therefore removed from the denominator).

To determine the intensity of individual spots, the median was used to reduce the influence of the bounding box size determined by the spot detection algorithm (peripheral pixels in the bounding box would bias the mean toward lower values more so than they bias the median). However, mean and total spot intensities were also measured in the course of the analysis and follow the same patterns.

We have added the following clarification to the Methods (lines 418-423): "The median intensity was chosen rather than the total or mean intensity to reduce the influence of bounding box size and of peripheral pixels in the bounding box, though the three measures do not behave in qualitatively different ways. Where possible we report spot counts as our primary measure of global *sim* expression, as it is largely independent of fluorescence intensity and thus slightly more robust."

Ultimately, the measurements used are an approximation of the true *sim* expression across the embryo. Future improvements building on our methodology might include the addition of a nuclear marker in a third channel to enable single-cell measurements of *sim* expression as well as Fluorescence Correlation Spectroscopy (FCS) to calibrate from fluorescence intensity to molecular concentrations. However, such additions to our already sophisticated approach are not required to support the claims in this paper and are thus best left to future work.

4. The Hairless hypomorph and *Twi* over expression data in figure 4 are presented as number of *sim* spots. The results from these experiments are used to conclude the state-dependent inactivation model best describes the experimental *sim* data. However, the models do not contain a spatial component and are actually giving the levels of *sim* expression, not the number of *sim* spots. *sim* expression levels need to be shown in figure 4 instead of number of *sim* spots.

As detailed above, both the number of spots and the total spot intensity are measures of *sim* expression, with the former being slightly more robust and the latter being slightly more appropriate for modelling. As one would expect, the two are almost perfectly linearly correlated (see figure below) and can thus be used interchangeably (given that their absolute scale is arbitrary).

We have added the following sentence in the Methods (lines 520-524): “We here used total *sim* expression rather than spot count, as the latter is a discrete measure and thus less suited for ODE fitting. Both spot count and total spot intensity are highly correlated and thus both represent robust measures of *sim* expression.”

5. In lines 515-519, I don't understand why the data was rescaled. The model should be fit to the data, so it should match the scale of the data.

The justification for this is given in those very lines. To re-iterate: both the model and the fluorescence intensity data are on arbitrary scales. There is no "scale of the data" that the model "should match". Linear rescaling by a fixed factor on the model or the data or both makes no difference whatsoever, except that bringing them into a scale of neither tiny nor large numbers is computationally convenient.

6. In general, I have concerns with the implementation of the modeling. I appreciate that A and B are not assigned molecular identities to avoid bias, however I think it needs to be clearly stated whether the models are trying to capture activation of already translated proteins (like a phosphorylation event as a molecular example) or if they are trying to capture activation of genes and subsequent protein translation. Both approaches have their merits, but would require different differential equations and different interpretations.

We have clarified in the text that we are indeed not considering regulation via transcription and translation of other genes, which would be unlikely at the time scales involved. Instead, we expect the motifs to be implemented by protein-protein interactions leading to conformational changes and potentially post-translational modifications. This is a very reasonable expectation given that interactions of this kind are well-known to take place extensively in enhancer-promoter regulation, including in the case of NICD, see for example the review by Bray S. (2016) Nat. Rev Mole Cell Bio.

We have added the following sentence (lines 183-185): "In accordance with the relatively short time scales involved we chose equations that best reflect an implementation of the motifs by protein-protein interactions rather than by gene regulatory networks."

7. As far as I know, NICD acts as a transcription factor and there is no reason to assume that NICD doesn't act directly on the sim enhancer. Similarly, it is not expected that the NICD would act on other proteins to activate them via something like phosphorylation (however it could form complexes, which would require refining the differential equations). Justification for why it is assumed that NICD acts through an intermediary, A, is necessary, and if A is treated as a transcription factor that is being transcribed due to activation by NICD, the differential equations need to be adjusted.

NICD is not a transcription factor but rather a co-activator; it requires Su(H) to bind DNA. This is text book information and already stated in the text (lines 61-63).

8. Why are $(1-x)$ terms (lines 480, 483, 486; equations 5, 7, 9) included in the production of A and B? This is not included in the models in figure 3, but implies that additional forms of A and B are considered and this value represents A and B subtracted from a total A or total B which has been normalized to one. This assumption needs to be justified, as I would expect total A and B could be changing at such a dynamic time during embryogenesis. Also, this model strongly suggests A exists in an inactivated form and is being activated by the NICD (as opposed to gene activation, where this model would not be appropriate).

Indeed, we do not assume production of A and B on the spot but rather their activation, consistent with protein-protein interactions rather than gene expression. Omitting the inactive form in the motif visualization for the sake of simplicity (as in figure 3) is common practice for diagrams of such signalling pathways, as illustrated by the fact that Ferrell (2016) used the same representations (countless other examples can be found in textbooks, research papers and reviews on the topic).

Although we obviously cannot definitively exclude that the inactive forms might exhibit some hidden dynamics, the fact is that we did not need to assume such dynamics to explain our data. Furthermore, specific experiments such as late photo-activation (figure 2B) provide some evidence against hidden dynamics. Including such dynamics in the model would thus be pure speculation and would just add free parameters, as experimental screening for and quantification of potential hidden dynamics in every component that might be involved in adaptation is obviously far beyond the scope of this (or any) paper.

9. Instead of manipulating constants to mimic the effects of an attenuator or stabilizer on A, the attenuator and stabilizer should be included explicitly in the differential equations. The manipulation of constants implies the attenuator and stabilizer are at steady state, which may not be true at such a dynamic time during embryogenesis.

The previous point applies here as well: introducing additional dynamic parts to the model would only add free parameters that are unconstrained unless numerous additional experiments are performed (in this case an analysis of the endogenous dynamics of Twist and Hairless).

In summary regarding points 6 through 9, we would like to stress again that *by definition* it remains possible that the true molecular model is a more complicated circuit that just so happens to largely match our model's predictions within the domain of our observations. We clearly state this limitation in the discussion (lines 287-301).

10. It is stated that Twi is necessary for *sim* expression (line 241), however, in the opto-Notch system, *sim* is expressed in regions where Twi is not expressed. These statements are contradictory. As far as I know, Twi is a transcription factor, so other than binding in a complex, Twi would have no additional role as a stabilizer. If a complex is being formed, this should be modeled explicitly. I think this study would benefit from trying to model the activation of *sim* directly instead of assuming that *sim* is proportional to A, and to do so by writing their own differential equations instead of using the previously published differential equations that were modified.

In lines 239-243 we specifically stated "The transcription factor Twist is a suitable molecular tool to test the consequences of stabilisation of (A), as mutations in its binding sites in the *sim* enhancer cause a loss of *sim* expression, yet on its own Twist is not capable of activating *sim* expression (Falo-Sanjuan et al., 2019, Kasai et al. 2019)." Importantly, the loss of *wild-type* *sim* expression in the mesectoderm as a result of mutated Twist binding sites does not preclude that *sim* can be expressed at least transiently in response to a strong exogenous burst of NICD, which is what we observe. This is consistent with Twist's role as a stabilizer in our model. Indeed, adaptation may very well be the mechanism that explains Kasai et al.'s finding that Twist is required for wild-type *sim* expression.

In any case, for clarity we have modified this sentence (lines 242-246) to "The transcription factor Twist is a suitable molecular tool to test the consequences of stabilisation of (A), as mutations in its binding sites in the *sim* enhancer cause a loss of wild-type *sim* expression in the mesectoderm, yet on its own Twist is not capable of activating *sim* expression (Falo-Sanjuan et al., 2019, Kasai et al., 1998)."

As for the modelling, the reviewer correctly suggests that the molecular mechanism underlying Twist's role as a stabilizer is likely complexing within the enhancer-promoter region. However, as explained above, explicitly modelling this interaction would simply add free parameters to the model, among which one or more configurations would trivially reproduce the model's current output. Thus, modelling the interaction with Twist explicitly would only make sense if direct measurements of Twist indicate notable dynamics on the

time scale of our experiments. This is, once again, beyond the scope of this study and needs not be assumed to explain the observed sim dynamics.

Minor Concerns

1. In line 101 and 119 it is not clear in the main text that "dark" refers to the first time point. It would help to add a sentence stating this.

In line 101 it is actually dark as we are imaging only mCherry. In line 119 we explained in the figure legend what that means. It would disturb the flow of the text to explain in the main text how we have acquired the first snapshot.

2. Do the authors have an explanation for why ectopic sim expression is lost but the wild type pattern remains? Since adaptation appears to be occurring (at least in the ectopic domain), why doesn't sim turn off in all nuclei instead of restricting to the wild type pattern? I assume the response would be that Twi is stabilizing the system so it cannot adapt. Were the levels in the wild type sim domain looked at by themselves to see if adaptation is occurring there to some degree?

Yes, this is indeed our interpretation, as we also discuss in the text. We have not quantified whether adaptation takes place to a lesser degree in the mesectoderm. While this may be an interesting question, it is not required to support the findings of the paper and would not notably add to them.

3. At the end of movie EV4 the NICD is no longer enriched in nuclei. Is there an explanation for this?

As explained in the movie legend and methods, the NICD (mCherry) signal bleaches at the end of the movie. However, as requested in the previous revision, we address the issue of bleaching extensively in our analysis (see figure EV1) and demonstrate that the amount of NICD in the nucleus is stable (see figure 2D).

4. In lines 338-340 the sentence is unclear.

We changed this sentence (now lines 341-343) to "On the one hand, the presence of factors such as Twist allows the continuous expression of target genes by counteracting adaptation."

5. Does Twi overexpression turn sna on, and if so, why doesn't this block sim activation?

We did not look at sna expression under twist overexpression. The fact that sim activation proceeds in accordance with our model supports our claims independent of the various potential reasons for sna's non-interference.

Dear Stefano,

Thank you for the submission of your revised manuscript to The EMBO Journal and for your patience during the evaluation process. As you will see below, your article has been seen by referee #1, who now consider that you have properly dealt with all of his/her major concerns.

Before we can proceed with the acceptance of your study, the referee proposes minor text changes that you can easily incorporate into the final version of the manuscript. Additionally, the format of the references needs to be slightly modified: a maximum of 10 authors can be listed, after which you must use "et al."

Please let me know if you have any further questions regarding any of these points. Thank you again for giving us the chance to consider your manuscript for The EMBO Journal and congratulations on a successful publication!

I look forward to receiving the final version of your manuscript with these minor changes included.

Yours sincerely,

David del Alamo
Editor
The EMBO Journal

Please click on the link below to submit the revision online:

Link Not Available

Referee #1:

The manuscript is much improved and the reviewer supports publication. The authors should include a reference for the statement line 329: "Twist expression is restricted to the mesoderm and a few additional rows of cells immediately dorsally". The authors are also encouraged to temper the conclusion (last line) that they have shown that a "linear signal transduction system, which does not involve signal amplification or other forms of relay through cytoplasmic proteins," is acting. The modeling supports this working hypothesis but it has not been proven.

The authors performed the requested editorial changes.

Dear Dr. De Renzis,

I am pleased to inform you that your manuscript has been accepted for publication in the EMBO Journal.

If you have any questions, please do not hesitate to call or email the Editorial Office. Thank you for your contribution to The EMBO Journal.

Yours sincerely,

David del Alamo
Editor
The EMBO Journal

Please note that it is EMBO Journal policy for the transcript of the editorial process (containing referee reports and your response letter) to be published as an online supplement to each paper. If you do NOT want this, you will need to inform the Editorial Office via email immediately. More information is available here:

<https://www.embopress.org/page/journal/14602075/authorguide#transparentprocess>

Your manuscript will be processed for publication in the journal by EMBO Press. Manuscripts in the PDF and electronic editions of The EMBO Journal will be copy edited, and you will be provided with page proofs prior to publication. Please note that supplementary information is not included in the proofs.

Please note that you will be contacted by Wiley Author Services to complete licensing and payment information. The 'Page Charges Authorization Form' is available here:

https://www.embopress.org/pb-assets/embo-site/tej_apc.pdf

Should you be planning a Press Release on your article, please get in contact with embojournal@wiley.com as early as possible, in order to coordinate publication and release dates.

Corresponding Author Name: Stefano De Renzis

Manuscript Number: EMBOJ-2020-107245